# The kallikrein–kinin pathway as a mechanism for auto-control of brown adipose tissue activity

Marion Peyrou [1], Rubén Cereijo[1], Tania Quesada-López [1], Laura Campderrós[1], Aleix Gavaldà-Navarro [1], Laura Liñares-Pose [2], Elena Kaschina[3,4], Thomas Unger[5], Miguel López [2], Marta Giralt [1] & Francesc Villarroya [1]✉

Brown adipose tissue (BAT) is known to secrete regulatory factors in response to thermogenic stimuli. Components of the BAT secretome may exert local effects that contribute to BAT recruitment and activation. Here, we found that a thermogenic stimulus leads to enhanced secretion of kininogen (Kng) by BAT, owing to induction of kininogen 2 (*Kng2*) gene expression. Noradrenergic, cAMP-mediated signals induce KNG2 expression and release in brown adipocytes. Conversely, the expression of kinin receptors, that are activated by the Kng products bradykinin and [Des-Arg9]-bradykinin, are repressed by thermogenic activation of BAT in vivo and of brown adipocytes in vitro. Loss-of-function models for Kng (the circulating-Kng-deficient BN/Ka rat) and bradykinin (pharmacological inhibition of kinin receptors, kinin receptor-null mice) signaling were coincident in showing abnormal over-activation of BAT. Studies in vitro indicated that Kng and bradykinin exert repressive effects on brown adipocyte thermogenic activity by interfering the PKA/p38 MAPK pathway of control of *Ucp1* gene transcription, whereas impaired kinin receptor expression enhances it. Our findings identify the kallikrein–kinin system as a relevant component of BAT thermogenic regulation that provides auto-regulatory inhibitory signaling to BAT.

[1] Departament de Bioquímica i Biomedicina Molecular, Institut de Biomedicina, Universitat de Barcelona (IBUB) and CIBER Fisiopatología de la Obesidad y Nutrición, Avda Diagonal 643, 08028 Barcelona, Spain. [2] Grupo NeurObesity, Departamento de Fisiología, CiMUS, Universidade de Santiago de Compostela-Instituto de Investigación Sanitaria and CIBER Fisiopatología de la Obesidad y Nutrición (CIBERobn), 15706 Santiago de Compostela, Spain. [3] Charité Universitäts medizin Berlin, Corporate member of Freie Universität Berlin, Humboldt-Universitätzu Berlin, and Berlin Institute of Health, Institute of Pharmacology, Center for Cardiovascular Research (CCR), Berlin, Germany. [4] DZHK (German Centre for Cardiovascular Research), Partner site Berlin, Berlin, Germany. [5] CARIM—School for Cardiovascular Diseases, Maastricht University, Maastricht, The Netherlands. ✉email: fvgombau@gmail.com

Obesity and type II diabetes are reaching pandemic proportions worldwide. Understanding the mechanisms underlying disturbed glucose and lipid metabolism is thus of great interest for efforts to identify new therapeutic targets for the treatment of these disorders.

Experimental studies have shown that impaired brown adipose tissue (BAT) activity is associated with obesity and insulin resistance. Conversely, highly active BAT is associated with a healthy metabolic phenotype that is generally attributed to the capacity of BAT to dissipate metabolic energy as heat through uncoupling protein 1 (UCP1)-mediated uncoupled mitochondrial oxidation[1]. In rodents, in addition to anatomically defined BAT sites that arise during ontogeny, brown adipocytes of distinct cellular origin, termed beige or brite, appear in white adipose tissue (WAT) depots after persistent thermogenic activation in the organism, a phenomenon referred to as browning or beiging of WAT[2]. This type of brown adipocyte appears to be the one that predominates in adult humans[3]. Genetic studies in rodents attribute an especially relevant role of the beiging process to protection against obesity[4].

Recent findings suggest that the biological function of BAT might not be restricted to the unique function of energy consumption, and reports based on experimental BAT transplantation point to a role for this tissue as a source of endocrine signaling and release the so-called brown adipokines or batokines[5,6]. Given the very distinct functions of WAT and BAT in energy metabolism, it would be reasonable to expect that the profile of batokines released by BAT would be different from adipokines released by WAT. Our laboratory recently conducted a study aimed at discovering new potential molecules secreted by BAT, based on microarray and RNAseq data in BAT combined with bioinformatic prediction of secretability. Two candidate genes—Cxcl14, a recently characterized chemokine released by BAT[7], and Kng (kininogen)—were identified. The kallikrein–kinin system, of which Kng is a part, is a complex hormonal signaling system with reported involvement in inflammation, blood pressure control, coagulation, and pain[8]. Because of alternative splicing, Kng encodes two different proteins: a high-molecular-weight Kng (HMWK) and a low-molecular-weight Kng (LMWK)[9]. In the human genome, only one gene (KNG1) has been described to date, whereas the mouse genome possesses a second gene, Kng2, in addition to Kng1[10]. These two murine genes have a sequence identity of ~89%. The liver is considered the main site of KNG proteins production. Once HMWK is produced and released into the bloodstream, it is targeted by plasmatic kallikrein protease, yielding the active peptide bradykinin. In contrast, LMWK is cleaved by tissue kallikrein, releasing the active kallidin into the blood[11]. Bradykinin and kallidin are considered to act as vasodilators and target many cell types by activating the kinin B2 (B2) receptor, which is constitutively expressed in cells. Bradykinin and kallidin are ultimately cleaved again to produce the peptides, [Des-Arg9]-bradykinin/kallidin; these peptides act on the kinin B1 (B1) receptor, whose expression is induced by distinct stimuli, including inflammation[8]. A summary of these pathways is shown in Fig. 1a.

Very few reports to date have indicated a role for the kinin system in systemic metabolic regulation and adipose tissue biology. One group proposed that a B1 receptor deficiency leads to leptin hypersensitivity and resistance to obesity[12]. A subsequent study from the same group suggested that kinin B1 and B2 receptor deficiencies improve glucose tolerance and have a protective role against high-fat-diet-induced obesity in mice[13]. To date, however, no studies describing a role for Kng in BAT functions or secretory properties have been reported. Here, we identify KNG2 as a component of thermogenic stimulus-induced BAT adaptations and describe a key role for the kallikrein–kinin system in adipose tissue plasticity in response to thermogenic challenges.

## Results

**Kng2 is the preferential Kng gene expressed in BAT**. To identify new brown adipokines, we performed a bioinformatic analysis of transcripts encoding potentially secreted proteins that are differentially expressed in mouse BAT-versus-WAT and cold-stimulated BAT versus BAT from mice at a thermoneutral temperature. This analysis identified two candidate genes, Cxcl14, further confirmed as a functional brown adipokine that targets immune cells[7], and Kng1. The identification of Kng1 as a gene that is preferentially expressed in interscapular BAT (iBAT) relative to white fat depots and induced in iBAT in response to thermogenic challenge of mice is consistent with a previous microarray-based report[14]. However, attempts to validate regulation of the Kng1 transcript in iBAT in response to cold by specifically measuring Kng1 transcript levels yielded results inconsistent with omics-based data. This prompted us to explore whether the presence of the closely related, highly homologous, gene Kng2 might have influenced the initial omics-based identification of Kng1 as a regulated gene.

Using primers designed to allow specific measurement (see Supplementary Methods) of Kng1 and Kng2 transcripts abundance, in both cases distinguishing between HMWK- and LMWK-encoding transcripts, we found that Kng1 transcripts were indeed undetectable in iBAT from Swiss mice and showed minor, but detectable, expression in iWAT (Fig. 1b, left). This contrasted with the liver, where Kng1 was highly expressed. However, Kng2 transcripts, especially LMWK, were markedly expressed in iBAT. In fact, the relative abundance of the LMWK Kng2 form in iBAT was in the range of that in the liver, the main Kng-producing tissue, whereas the level of the HMWK Kng2 transcript in iBAT was approximately one-third of that in liver (Fig. 1b, left).

**BAT activation and WAT browning increase KNG2 expression**. We found that Kng1 transcript expression remained undetectable in iBAT of cold-exposed Swiss mice, whereas cold dramatically induced the expression of both HMWK and LMWK Kng2 transcripts (Fig. 1b, right). Although expression of the HMWK Kng2 transcript was not induced by cold exposure in iWAT, expression of the LMWK Kng2 transcript increased (Fig. 1b, right). These effects occurred specifically in adipose tissues, as there was no evidence for a cold-induced increase in Kng transcript abundance in the liver (Fig. 1b, right), muscle, heart, or intestine (Supplementary Fig. 1). The levels of KNG2 protein in iBAT and iWAT from cold-exposed mice were significantly upregulated, consistently with Kng2 transcript levels (Fig. 1c). These data establish that Kng2 is the gene that is actually regulated by a thermogenic stimulus in iBAT. The original identification of Kng1 as a regulated transcript in omics-based data was thus likely attributable to the very high sequence similarity between the two genes and an inability of hybridization-based quantification in microarray assays to discriminate between them. Notably, a study by Fitzgibbons et al. previously identified Kng2 as being preferentially expressed in iBAT[15].

We next investigated the effect of cold exposure on plasma levels of circulating Kng (HMWK type) and found a significant cold-induced increase in KNG2 levels, but not KNG1 levels (Fig. 1d). This confirms the preferential sensitivity of KNG2 protein synthesis to cold challenge as well as the systemic impact of thermogenic activation of BAT on the Kng system.

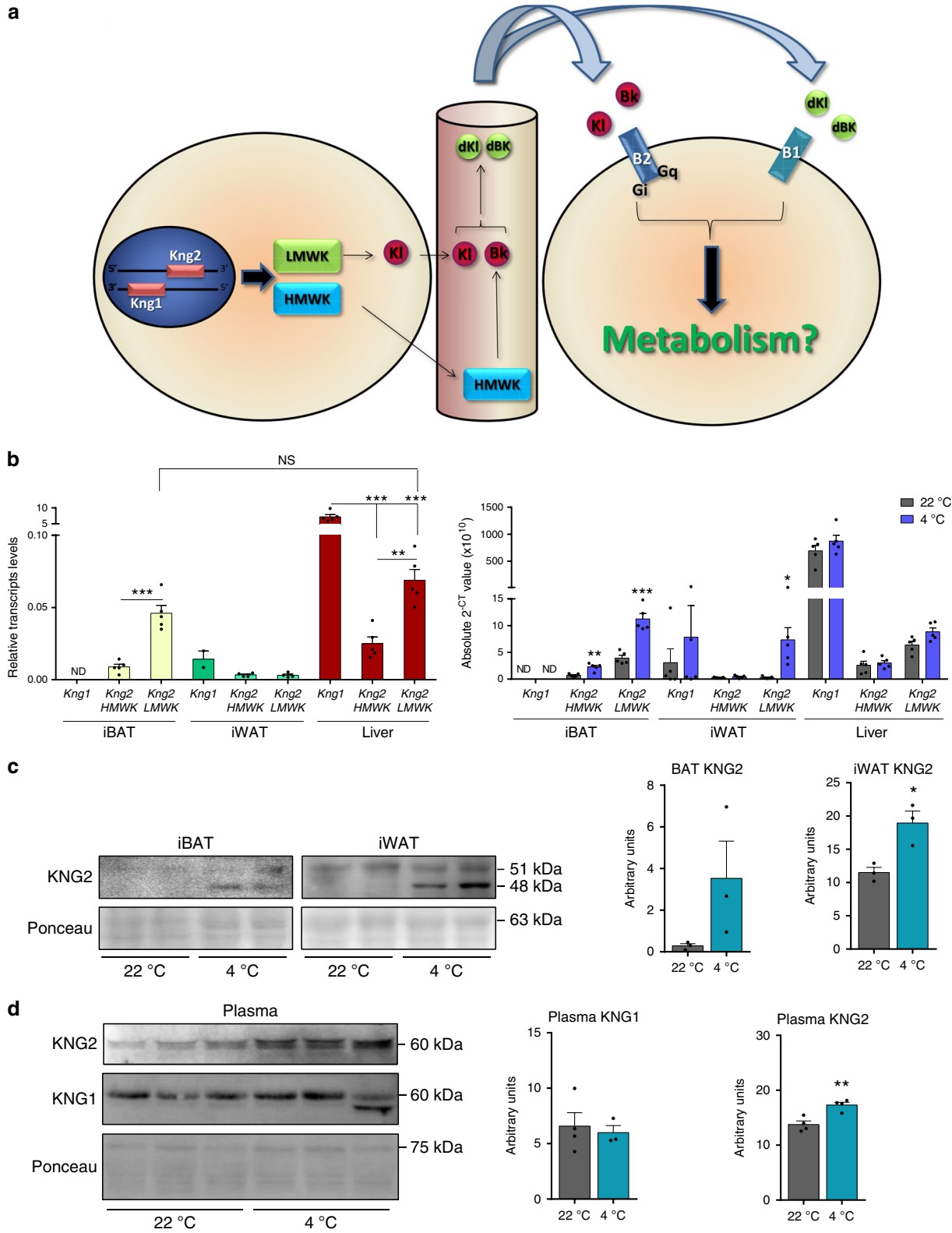

**Fig. 1 Differential expression of *Kng* genes in tissues. a** Representation of the Kng system. LMWK, low molecular weight kininogen; HMWK, high molecular weight kininogen; Kl, kallidin; Bk, bradykinin; Dkl, [Des-Arg9]-kallidin; Dbk, [Des-Arg9]-bradykinin; B2, kinin B2 receptor; B1, kinin B1 receptor. **b** mRNA expression of the different *Kngs* in iBAT, iWAT and liver of 2 months old Swiss mice (*n* = 5 animals; except for iWAT *Kng2* HMWK and LMWK left graph where *n* = 4). mRNA expression of *Kngs* in fat depots and liver of 2 months old Swiss mice exposed to cold or control room temperature for 1 week (*n* = 5 animals). **c** KNG2 protein expression in iBAT and iWAT of mice exposed to RT or 4 °C (*n* = 3 animals). **d** KNG1 and -2 protein expression in plasma of WT mice exposed to cold for 1 week (*n* = 4 animals; except for KNG1 4 °C where *n* = 3). ND, not detectable; NS, not significant. Data are presented as means ± s.e.m. (bars). *$P < 0.05$, **$P < 0.01$, and ***$P < 0.001$ versus corresponding controls. *P*-values determined by two ways ANOVA with Tukey's post hoc test (**b** left) and two-tailed unpaired Student's *t*-test (**b** right, **c**, **d**). Source data are provided as a Source data file.

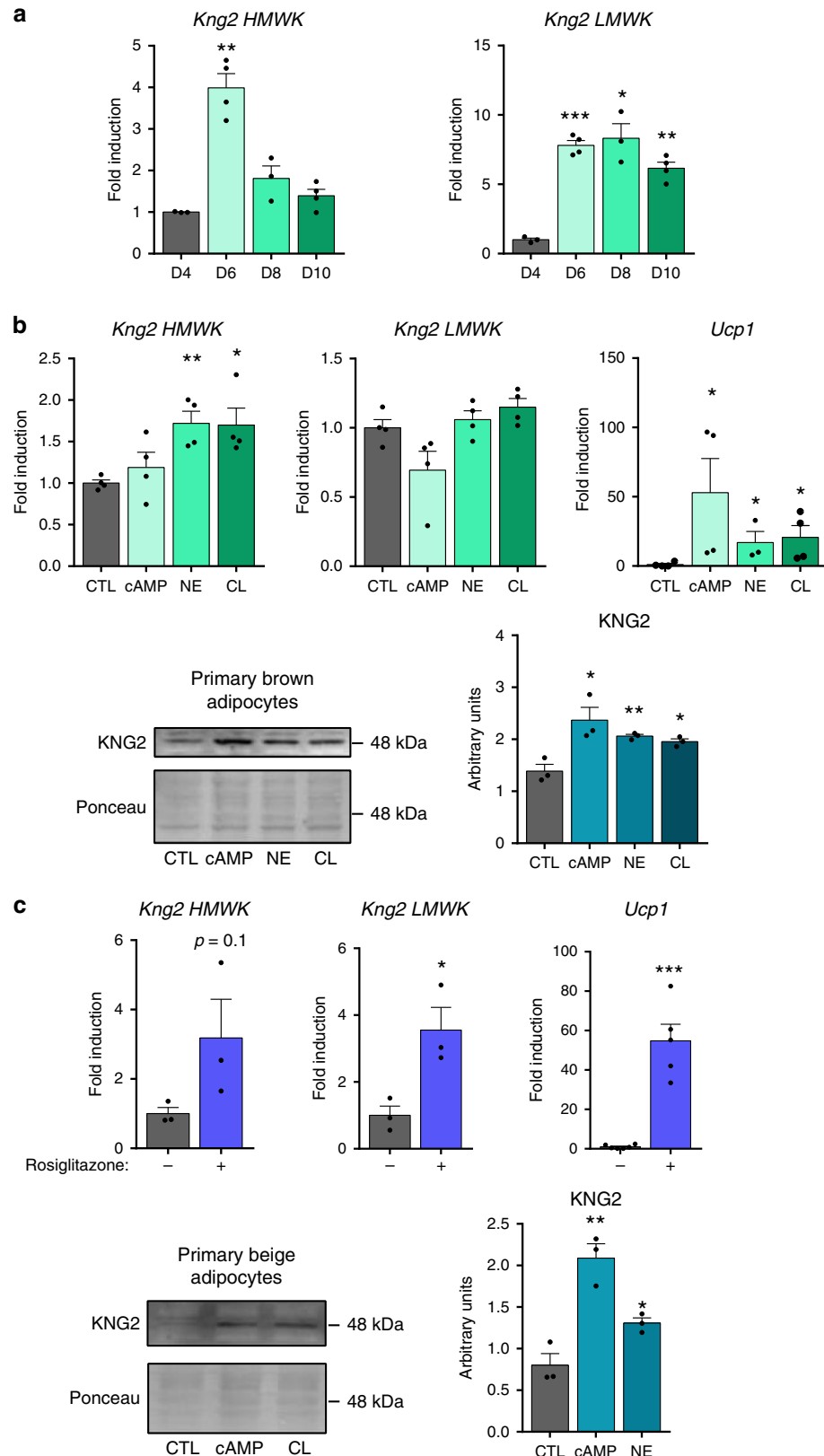

**Noradrenergic stimuli induce *Kng2* in brown adipocytes**. We then investigated whether thermogenically induced expression and release of KNG2 in BAT represents a cell-autonomous response of brown adipocytes to classic adrenergically mediated thermogenic stimulation. First, we found that HMWK *Kng2* mRNA was induced during brown adipocyte differentiation in vitro (mostly at early stages), whereas LMWK *Kng2* mRNA expression was dramatically increased in association with differentiation (Fig. 2a). We found that both norepinephrine and the β3-specific agonist CL316243 significantly increased HMWK *Kng2* mRNA levels (Fig. 2b). This resulted in an increase in KNG2 protein levels in brown adipocytes in response to

**Fig. 2 *Kng*2 regulation in murine primary brown and beige adipocytes. a** mRNA expression of the two *Kng*2 variants during primary brown adipocyte differentiation ($n = 3$ independent cell culture experiments for D4 and D8 and $n = 4$ for D6 and D10). D4–8, day 4–8 of differentiation; D10, day 10, fully differentiated adipocytes. **b** *Kng*2 mRNA ($n = 4$ independent cell culture experiments) and protein expression ($n = 3$ independent cell culture experiments) in primary brown adipocytes after stimulation with β-adrenergic activators. *Ucp1* is shown as a positive control for effects of drugs on cells. **c** *Kng*2 mRNA ($n = 3$ independent cell culture experiments for *Kng*2 HMWK and LMWK; $n = 5$ independent cell culture experiments for *Ucp1* −Rosi; $n = 6$ independent cell culture experiments for *Ucp1* +Rosi) and protein expression ($n = 3$ independent cell culture experiments) in primary inguinal white adipocytes after stimulation with rosiglitazone or β-adrenergic activators. *Ucp1* is shown as a positive control for the browning effect of rosiglitazone. Data are presented as means ± s.e.m. (bars). *$P < 0.05$, **$P < 0.01$, and ***$P < 0.001$ versus corresponding controls. *P*-values determined by one way ANOVA with Tukey's post hoc test (**a**–**c** down) and two-tailed unpaired Student's *t*-test (**c** up). Source data are provided as a Source data file.

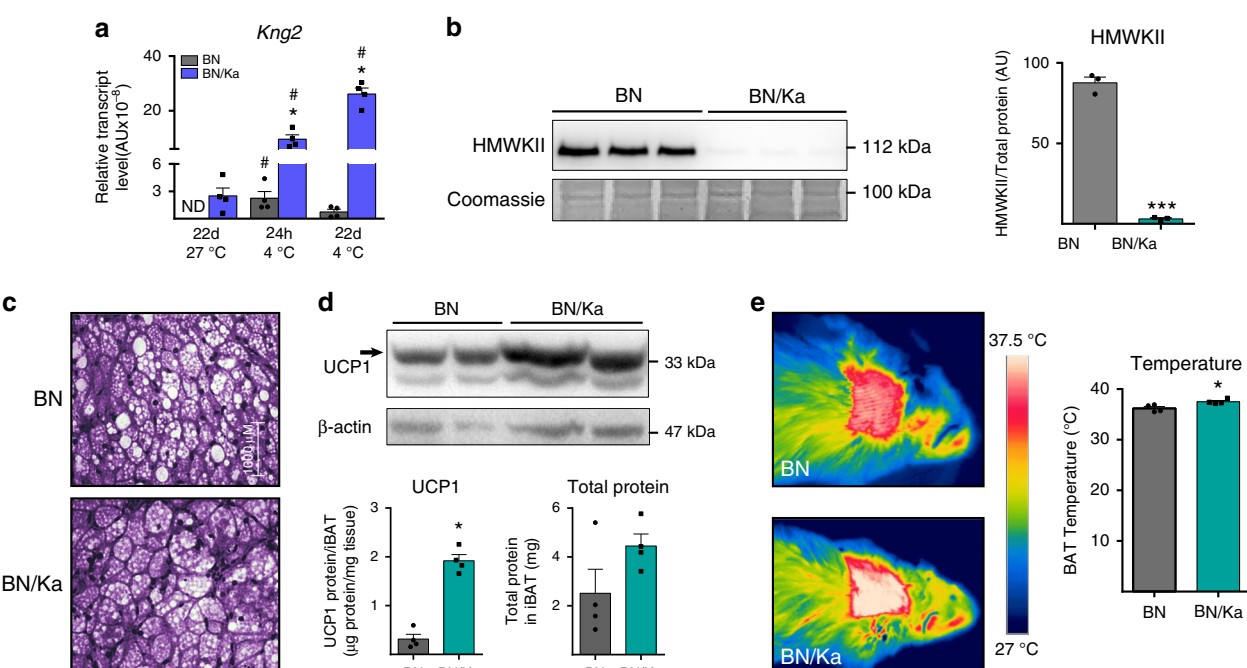

**Fig. 3 Regulation of *Kng*2 expression and BAT activity in BN/Ka rats. a** mRNA expression of K-kininogen (*Kng*2) in iBAT from 12 weeks old BN and BN/Ka rats at thermoneutrality (27 °C) and after 24 h or 22 days exposure to cold (4 °C) ($n = 4$ animals). **b** HMW-KNG2 protein expression in plasma and quantification in BN and BN/Ka rats under thermoneutral conditions ($n = 3$ animals). **c** Optical microscopy of H&E-stained iBAT from BN and BN/Ka rats under thermoneutral conditions ($n = 4$ animals). **d** UCP1 protein expression and quantification of UCP1 protein levels, expressed per tissue and total protein (below), in iBAT from BN and BN/Ka rats under thermoneutral conditions ($n = 4$ animals). **e** Representative infrared thermographies (left) and quantification of iBAT temperature (right) in BN and BN/Ka rats under thermoneutral conditions ($n = 4$ animals). ND, not detectable. Data are presented as means ± s.e.m. (bars). *$P < 0.05$ and ***$P < 0.001$ versus BN; #$P < 0.05$ versus thermoneutrality. *P*-values determined by two-tailed unpaired Student's *t*-test (for genotypes) and two ways ANOVA with Tukey's post hoc test (for temperature) (**a**) and two-tailed unpaired Student's *t*-test (**b**, **d**, **e**). Source data are provided as a Source data file.

norepinephrine, CL316243 and cAMP treatments, all of which elicited enhanced thermogenic activation, as shown by the induction of *Ucp1* transcript levels (Fig. 2b). In primary adipocyte cultures from iWAT precursors, induction of beiging by rosiglitazone (see massive induction of *Ucp1* gene transcript levels) also resulted in a tendency of HMWK *Kng*2 transcripts to increase and caused a significant induction of LMWK *Kng*2 transcript levels (Fig. 2c, top). Moreover, in differentiated beige adipocytes, CL316243 and cAMP increased KNG2 protein levels (Fig. 2c, bottom).

**Rats devoid of circulating Kng have over-activated BAT.** As a first approach for determining the role of Kng in vivo, we analyzed the phenotype of Brown Norway Katholiek (BN/Ka) rats in relation to systemic metabolism, thermoregulation, and BAT function. BN/Ka rats harbor a spontaneous point mutation of alanine to threonine at residue 163 in the KNG2 protein that leads to a defect in secretion of HMWK and a consequent deficiency of plasma Kng[16,17]. Control Brown Norway (BN) rats

showed enhanced *Kng*2 expression in iBAT in response to short- and long-term cold, similar to mice (Fig. 3a). As expected, we found that HMW-KNG2 protein was almost undetectable in the blood of BN/Ka rats (Fig. 3b). BN/Ka rats showed no gross alterations in body weight or food intake. They also showed no changes in insulin or pro-inflammatory cytokines, as previously reported[18]; the only change in metabolic parameters detected was a mild reduction in glycemia (Supplementary Table 1). Liver and WAT mass (estimated from the epididymal depot) were unaltered in BN/Ka rats, whereas the iBAT depot was significantly larger (Supplementary Table 1). A histological examination revealed that this enlargement was not attributable to fat accumulation (Fig. 3c). Both relative UCP1 protein levels and absolute UCP1 levels per iBAT (indicative of thermogenic potential) were significantly increased in iBAT from BN/Ka rats (Fig. 3d). That increased recruitment of iBAT in BN/Ka rats had functional consequences and is demonstrated by measurements of local temperature at the iBAT skin surface region, which was significantly increased (Fig. 3e). In summary, the deficiency in

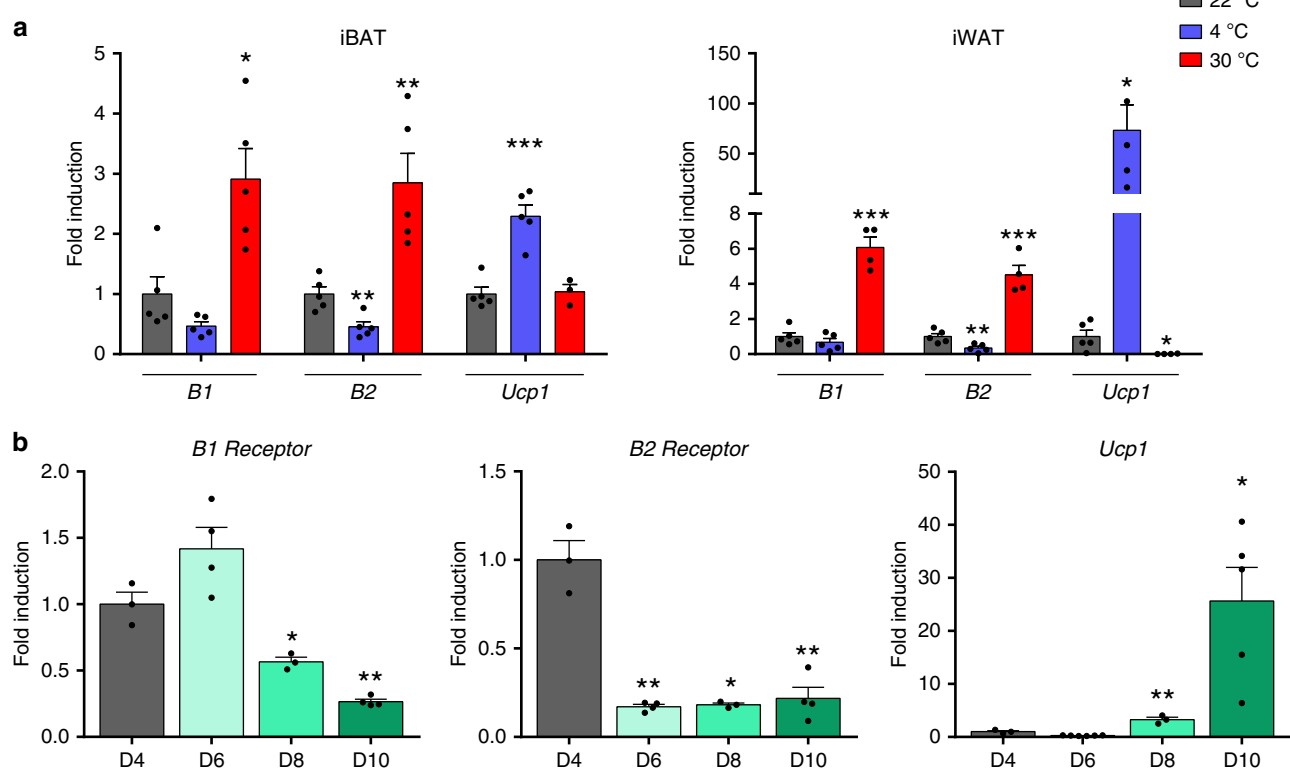

**Fig. 4 _B1_ and _B2_ receptor expression. a** _B1_ and _B2_ receptor mRNA expression in iBAT and iWAT of mice exposed to cold or thermoneutrality for 1 week (_n_ = 5 animals except for iBAT _Ucp1_ 30 °C where _n_ = 3 and iWAT 30 °C all genes where _n_ = 4). _Ucp1_ is shown as a control for the efficiency of thermogenic challenges. **b** mRNA expression of _B1_ and _B2_ receptors during differentiation (D4–8) and in fully differentiated brown adipocytes (D10) (_n_ = 3 independent cell culture experiments for D4 and D8 and _n_ = 4 for D6 and D10 for _B1_ and _B2_ genes; _n_ = 3 for D4 and D8 and _n_ = 6 for D6 and _n_ = 5 for D10 for _Ucp1_ gene). _Ucp1_ is shown as a positive control for differentiation. Data are presented as means ± s.e.m. (bars). *_P_ < 0.05, **_P_ < 0.01, and ***_P_ < 0.001 versus corresponding controls; one way ANOVA with Tukey's post hoc test. Source data are provided as a Source data file.

KNG2 protein in BN/Ka rats was associated to some extent with iBAT thermogenic activation. Notably, however, rats (but not mice or humans) harbor two additional genes encoding Kng-related proteins, T-Kng-I and T-Kng-II, which encode proteins with functions that are similar to but not completely overlapping with those of HMW-KNG2 and LMW-KNG2, encoded by the rat _Kng2_ gene[19]. We found that the deficiency in KNG2 secretion in BN/Ka rats was accompanied by a dramatic induction of T-Kng-I and T-Kng-II transcripts in iBAT (Supplementary Fig. 2). Thus, the possibility that T-Kng–mediated compensatory processes tone down the impact of a KNG2 deficiency in the BN/Ka rat cannot be ruled out. In any case, results obtained using the BN/Ka model suggest that the most sensitive target of a KNG2 deficiency is BAT itself.

**Effects of thermogenic challenges on _B1_ and _B2_ expression.** Considering that _Kng_ gene expression results in the release of HMWK and the bioactive peptides kallidin and bradykinin, which target cells through the kinin receptors B1 and B2, we analyzed the expression of these receptors in mouse iBAT as part of an exploration of putative local (autocrine/paracrine) effects of thermogenically induced _Kng2_ gene expression.

We found that expression of _B1_ and _B2_ genes was inversely regulated in accordance with the extent of thermogenic challenge in mice (Fig. 4a), such that cold exposure (4 °C, 1 week) caused a general trend to decrease _B1_ and _B2_ expression in iBAT, and exposure to 30 °C for 1 week, which suppresses thermogenically induced BAT activity, resulted in a dramatic induction of _B1_ and _B2_ gene expression. This response was opposite that of _Ucp1_ mRNA, used as a marker for the extent of thermogenic induction.

A similar behavior was found in iWAT, in which cold-induced promotion of browning tended to decrease _B1_ and _B2_ mRNA expression, whereas suppression of browning by exposure to 30 °C (which essentially abrogated _Ucp1_ mRNA expression) resulted in strong induction of _B1_ and _B2_ transcripts (Fig. 4a). We also found that expression of _B1_ and _B2_ receptor genes was significantly downregulated in association with differentiation of brown adipocytes. _B1_ and _B2_ transcripts were reciprocally regulated relative to those of _Ucp1_ throughout the time-course of brown adipocyte differentiation (Fig. 4b).

The transcript levels of other components of the kinin system, such as plasma kallikrein b1 (_Klkb1_), kininase I (_Klk1_) and angiotensin-converting enzyme (_Ace_) were not affected significantly by environmental thermal challenges (Supplementary Fig. 3).

**Pharmacologic inhibition of B1/B2 receptors overactivates BAT.** The fact that thermogenic-mediated induction of _Kng2_ is accompanied by repressive effects on _B1_ and _B2_ receptor expression raises the question of what effect inhibiting B1 and B2 receptor activity has on the responsiveness to a thermogenic stimulus. To address this question, we implanted mini-pumps containing a cocktail of B1 and B2 receptors antagonists in the subcutaneous dorsal area of mice, just over the interscapular BAT depot, and exposed mice to 4 °C for 1 week. Neither systemic metabolic and circulating profiles (Supplementary Table 2), nor iWAT parameters (Supplementary Fig. 4) were altered by this treatment. However, we found that, in mice treated with antagonists, the size of iBAT and the diameters of cells were smaller (Fig. 5a), the percentage of protein content in the tissue was higher, and the total amount of protein remained the same,

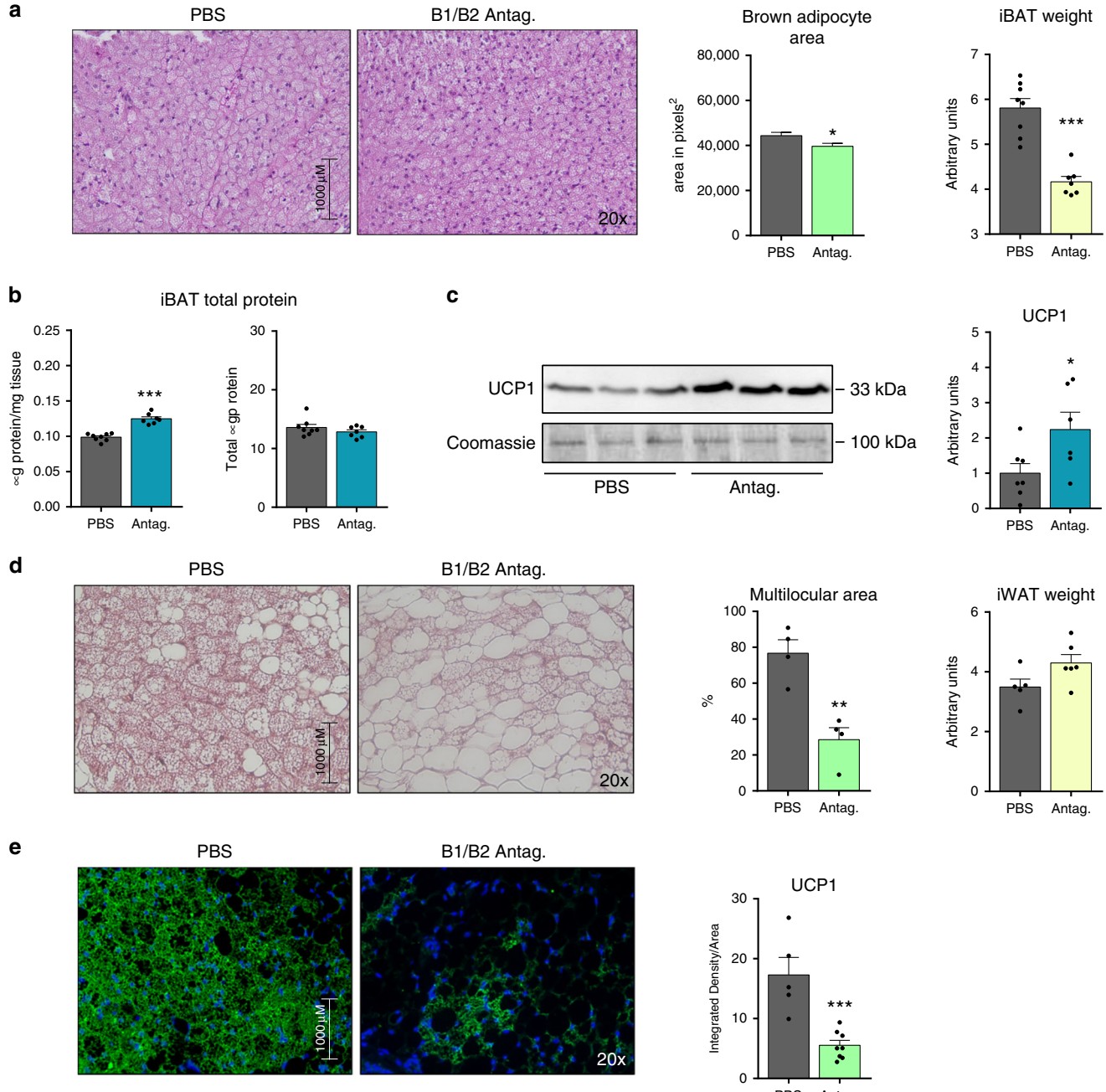

**Fig. 5 Effects of local pharmacological inhibition of B1/B2 receptors at iBAT and iWAT. a** H&E-stained iBAT histological sections and quantification (at least 3 different pictures) of cell size in 2 months old WT C57BL/6J mice exposed to 4 °C for 1 week and then implanted in the back with mini-pumps delivering either PBS or a cocktail of B1 and B2 antagonists; iBAT weight normalized to tibia length ($n = 8$ animals for PBS and $n = 7$ animals for antagonists). **b** Quantification of total protein per milligram tissue and in the entire iBAT depot ($n = 8$ animals for PBS and $n = 7$ animals for antagonists). **c** UCP1 protein expression in iBAT, with quantification ($n = 7$ animals for PBS and $n = 6$ animals for antagonists). **d** H&E-stained iWAT histological sections and quantification of browning area (at least three different pictures) in 2 months old WT C57BL/6 J mice exposed to 4 °C for 1 week and implanted in the leg with mini-pumps delivering either PBS or a cocktail of B1 and B2 antagonists; iWAT weight normalized to tibia length ($n = 5$ animals for PBS and $n = 6$ animals for antagonists). **e** Immunofluorescence detection of UCP1 protein in iWAT histological sections, with quantification (at least three different pictures). Data are presented as means ± s.e.m. (bars). *$P < 0.05$, **$P < 0.01$, and ***$P < 0.001$ versus corresponding controls; two-tailed unpaired Student's $t$-test. **a**–**c**) $n = 7$; **d**, **e**) $n = 6$. Source data are provided as a Source data file.

indicating that the reduction in the size of iBAT was attributable to a loss of fat and not protein (Fig. 5b). UCP1 protein levels in iBAT were increased in animals treated with the antagonist cocktail (Fig. 5c). Taken together, these data indicate that activation of BAT by cold exposure was more intense in antagonist-treated mice.

The above data suggest that our B1/B2-antagonist experimental setting may have exerted mostly local effects upon iBAT. To test this, we performed an additional experiment in which mini-pumps delivering B1/B2-antagonists were implanted locally adjacent to an iWAT depot site, and subsequently exposed mice to 4 °C for 1 week. This treatment did not alter systemic

metabolic parameters, such as glycemia and triglyceridemia, or iBAT parameters, such as iBAT weight or relative and total UCP1 protein levels (Supplementary Fig. 5). However, local infusion of B1/B2 receptor antagonists dramatically repressed the browning of iWAT in response to cold, as evidenced by impaired development of multilocular beige adipocytes (Fig. 5d) and reduced expression of UCP1 (Fig. 5e). From these observations, we conclude that local inhibition of B1/B2 receptors has reciprocal effects on thermogenic activation of BAT and browning of iWAT.

**Effect of B1/B2 receptor KO on response to thermal challenge**. Given the strong regulation of *Kng2* gene expression and kinin receptors in BAT and WAT (determined by thermogenic requirements and the extent of browning), and the differential effects of the kininogen system on BAT and iWAT, we undertook a study of the role of the kallikrein–kinin pathway using mice with targeted suppression of both *B1* and *B2* kinin receptor genes (B1B2R-KO mice)[20].

Under basal, room temperature (22 °C) conditions, these mice showed no gross alterations in growth, body weight or food intake, and their main biochemical and hormonal parameter profiles were unaltered (Supplementary Table 3). This is in agreement with previous data in this mouse model[21,22], but in contrast with a report using an independently-generated B1B2R-KO mouse that found low leptin and insulin levels[13]. We challenged two group of mice with two opposite experimental settings: (1) 1-week exposure to 4 °C, a thermogenic stimulus designed to promote WAT browning; or (2) 1-week exposure to a thermoneutral temperature (30 °C), which induces the whitening of BAT and WAT. B1B2R-KO mice exposed to 4 °C were not dramatically intolerant to cold, but general parameters related to the physiological response to cold environment were remarkably altered. The increase in food intake, an adaptive response to cold that occurs in wild type (WT) mice, was significantly reduced in B1B2R-KO mice, whereas the mass of inguinal, epididymal and mesenteric WAT was significantly increased in cold-exposed B1B2R-KO mice relative to WT mice (Supplementary Table 3). This latter observation reflects the fact that the reduction in WAT mass in WT mice in response to cold was impaired in B1B2R-KO mice. These results are consistent with the cold-induced reduction in circulating leptin levels in WT mice, but not in B1B2R-KO mice (Supplementary Table 3).

After suppression of the thermogenic stimulus (exposure to 30 °C), food consumption was reduced in WT mice and the mass of WAT depots was massively increased, consistent with impaired energy expenditure under these conditions (Supplementary Table 3). Such adaptations to thermogenic suppression were severely impaired in B1B2R-KO mice, which showed a generalized reduction in their capacity for thermoneutrality induced increases in WAT mass. This finding is consistent with the blunted adaptive reduction in food intake and much lower leptin levels in B1B2R-KO mice relative to WT mice (Supplementary Table 3).

**BAT overactivation is sustained in B1B2R-KO mice**. Exposure of B1B2R-KO mice to cold led to overactivation of BAT relative to cold-exposed WT mice, as evidenced by increased levels of UCP1 protein (Fig. 6b). This BAT overactivation was accompanied by increased levels of tyrosine hydroxylase (TH) protein, indicative of enhanced sympathetic innervation (Fig. 6b), although this was not reflected in massive changes in transcript levels (Supplementary Table 4). Conversely, B1B2R-KO mice showed impaired capacity to down-regulate BAT activity when transferred from 22 °C to thermoneutrality (30 °C), as evidenced

by impaired whitening of BAT based on cell morphology (Fig. 6a), maintenance of high levels of UCP1 protein (Fig. 6b), a trend toward higher levels of transcripts for thermogenesis-related genes (*Ucp1, Dio2, C/EBPb, Prdm16, Pgc1a*), and reduced expression of the whitening marker gene *Leptin* (Supplementary Table 4). The expression profile of transcripts for genes involved in fatty acid/glucose metabolism, angiogenesis and local immune status in iBAT from cold-exposed B1B2R-KO mice was essentially unaltered relative to that in WT mice (Supplementary Table 4). However, in a thermoneutral setting, marked changes in the transcript level profile were found in B1B2R-KO relative to WT mice. Specifically, we observed an overall increase in transcript levels for genes involved in fatty acid supply and oxidation (*Lpl, Cd36, Dgat, Lcad, Acox1*), and glucose uptake (*Glut4*) (Fig. 6c). Moreover, whereas expression of angiogenesis-related genes was unaltered, transcripts for *Tnfa*, a pro-inflammatory gene, were downregulated in iBAT from B1B2R-KO mice exposed to thermoneutral conditions, and *Clec10a* mRNA, corresponding to alternatively activated macrophages, was reciprocally upregulated (Fig. 6c). Collectively, these data confirm that the adaptive down-regulation of BAT activity in response to thermoneutrality is impaired in B1B2R-KO mice, and show that BAT is abnormally over-activated in B1B2R-KO mice, both under conditions of enhanced and reduced thermogenic requirements.

To directly investigate the functional capacity of BAT to induce adaptive thermogenesis in B1B2R-KO mice, we determined the thermal response of iBAT to the β3-adrenergic activator CL316243. Injection of CL316243 into the interscapular region caused a more intense induction of heat at iBAT locations in B1B2R-KO mice than in WT mice (Fig. 7a, b). We also measured the induction of $O_2$ consumption in response to CL316243 and found a significant increase in B1B2R-KO mice (Fig. 7c). These results confirm that *B1* and *B2* gene invalidation in mice results in greater thermogenic capacity in association with enhanced recruitment of BAT.

**WAT from B1B2R-KO mice does not adapt to thermal challenge**. We found distinct alterations in WAT from B1B2R-KO mice compared with those observed in BAT from these mice. Specifically, B1B2R-KO mice showed impaired WAT browning in response to cold, as evidenced by the reduced appearance of multilocular adipocytes in the iWAT depot (Fig. 8a) and reduced levels of UCP1 and TH proteins (Fig. 8b). In keeping with these findings, transcript levels of the thermogenesis-related genes, *Ucp1, Dio2* and *C/EBPb*, were downregulated in iWAT from cold-exposed B1B2R-KO mice relative to cold-exposed WT mice (Supplementary Table 5). Interestingly, the inducibility of *Kng2* gene expression in response to cold was strongly inhibited in B1B2R-KO mice (Supplementary Table 5). Moreover, transcript levels for genes encoding components of fatty acid and glucose metabolism, as well as those involved in angiogenesis and immune cell infiltration were largely unaltered in iWAT from B1B2R-KO mice relative to WT controls (Supplementary Table 5). On the other hand, suppression of thermogenic responses by exposure of mice to a thermoneutral temperature of 30 °C tended to be less effective in B1B2R-KO than WT mice, as evidenced by the smaller adipocytes in WAT from B1B2R-KO mice relative to those from WT mice. TH protein was also maintained at higher levels in B1B2R-KO mice relative to WT mice under thermoneutral conditions (Fig. 8b).

**The kallikrein–kinin system inhibits brown/beige adipocytes thermogenesis**. Having established that the thermogenic recruitment of iBAT and iWAT are altered following experimental manipulation of the kallikrein–kinin system in rodent

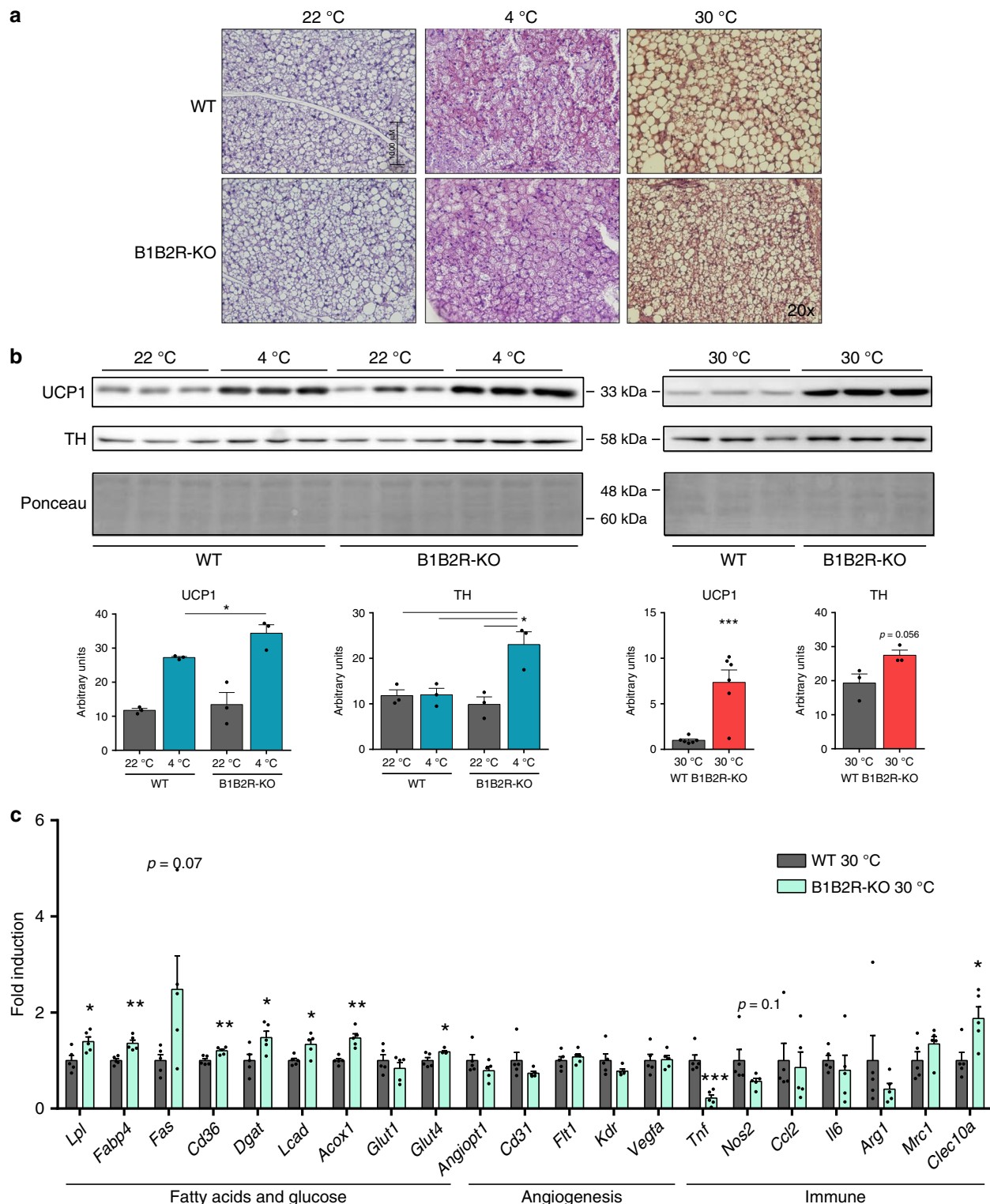

**Fig. 6 Effects of temperature challenges on BAT from B1B2R-KO mice. a** H&E-stained histological sections (at least 3 different pictures) of iBAT from 3 months old WT and B1B2R-KO exposed to 4 °C or 30 °C for 1 week. **b** Expression and quantification of UCP1 and TH protein ($n = 3$ animals except for UCP1 30 °C where $n = 6$). **c** mRNA expression of genes involved in fatty acid and glucose metabolism, angiogenesis, and the immune system ($n = 5$ animals). Data are presented as means ± s.e.m. (bars). *$P < 0.05$, **$P < 0.01$ and, ***$P < 0.001$ versus corresponding controls; two-tailed unpaired Student's $t$-test. Source data are provided as a Source data file.

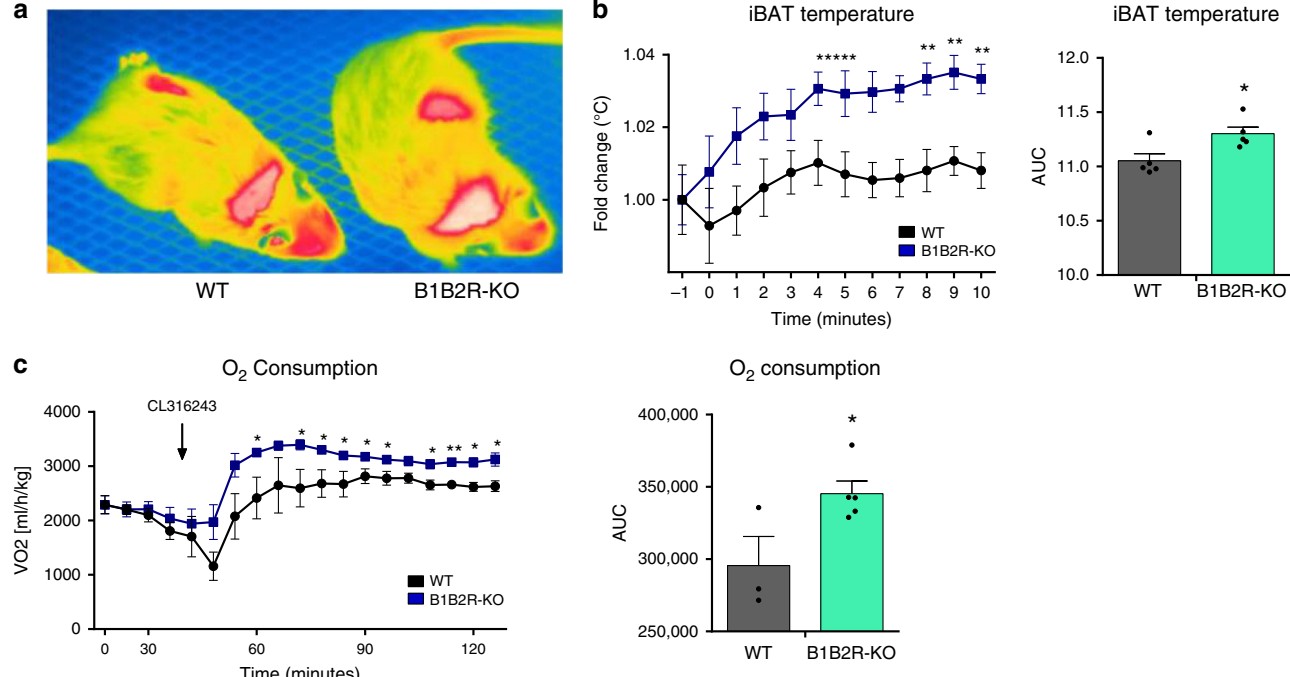

**Fig. 7 Effects of CL316243 injection on iBAT temperature and O$_2$ consumption. a** Thermographic image of a WT and B1B2R-KO mouse 4 min after subcutaneous injection of CL316243 in the area of iBAT. **b** Time course of temperature increase in iBAT after CL316243 injection ($n = 5$ animals for PBS and $n = 6$ animals for B1B2R-KO left graph; $n = 5$ animals for AUC). **c** Time course of O$_2$ consumption after CL316243 injection ($n = 3$ animals for PBS and $n = 5$ animals for B1B2R-KO both graphs). Data are presented as means ± s.e.m. (bars). *$P < 0.05$, **$P < 0.01$ versus corresponding controls; two-tailed unpaired Student's $t$-test. Source data are provided as a Source data file.

models, we investigated the cell-autonomous responsiveness of brown and beige adipocytes.

We found that the treatment of brown adipocytes with bradykinin and recombinant HMWK protein significantly repressed *Ucp1* transcript expression (Fig. 9a, left). Moreover, HMWK caused some reduction in the extent of brown adipocyte differentiation, as evidenced by reduced appearance of multi-locular lipid droplets in cells (Fig. 9a, right). We also found repressive effects of bradykinin and HMWK on lipolysis (glycerol release), which contrasted with the known inducing effects of cAMP (Fig. 9b). Brown adipocytes differentiating from precursors derived from iBAT of B1B2R-KO mice showed a mild increase in morphological differentiation compared to cells derived from WT mice (Fig. 9c, right), as well as an impaired capacity to down-regulate *Ucp1* transcript in response to bradykinin (Fig. 9c, left). Moreover, brown adipocytes from B1B2R-KO mice expressed higher levels of UCP1 protein both under basal and cAMP- and norepinephrine-induced conditions relative to WT cells (Fig. 9d). These data support an inhibitory role of kallikrein–kinin derived products, mediated by B1/B2 kinin receptors, in brown adipocytes.

The general adipogenic differentiation of precursor cells obtained from iWAT of B1B2R-KO mice was mildly increased relative to that of cells from WT mice, especially under beiging-inducing treatment (rosiglitazone) (Fig. 9e). In precursors from B1B2R-KO iWAT induced to beiging, the transcript levels of the general adipogenic factor PPARγ and the adipogenic marker FABP4 were significantly increased, and induction of *Ucp1* and *Cidea* transcripts was also observed (Fig. 9f). Accordingly, UCP1 protein levels were significantly upregulated in B1B2R-KO beige cells (Fig. 9g).

We analyzed how iWAT precursor cells differentiated to a preferential white phenotype (no rosiglitazone treatment, UCP1 expression around 10% of that in beige adipocytes) responded to

bradykinin; we found no repressive effects, but instead observed a significant induction of UCP1 protein levels (Fig. 9h). In contrast, when cells were driven to a beige phenotype (rosiglitazone treatment), bradykinin treatment reduced UCP1 protein levels (Figs. 9h and 9i), similarly to our previous observations in brown adipocytes.

Collectively, these findings confirm that the kallikrein–kinin system exerts repressive effects on the thermogenic activation of brown and beige adipocytes, whereas no such effect (and even the opposite effect) is seen in white adipocytes.

**BK inhibits the p38–MAPK pathway of thermogenic activation.** In order to ascertain the mechanisms through which bradykinin exerts repressive effects on brown adipocyte thermogenic activation, we first analyzed the potential involvement of G$_q$ and/or G$_i$-subtype G-protein-coupled receptors, which are potential targets of bradykinin action in other cell systems[23]. Treatment of brown adipocytes with a G$_q$ inhibitor did not alter the repressive action of bradykinin on *Ucp1* or *Cidea* transcript levels, as biomarkers of brown fat thermogenic activation. In contrast, G$_i$ inhibition suppressed the repressive effect of bradykinin on these transcript levels (Fig. 10a, left). Consistent with the findings shown previously, bradykinin did not repress thermogenesis-related *Ucp1* and *Cidea* transcript levels in white adipocytes, instead tended to increase them; moreover the G$_q$ and G$_i$ inhibitors did not have any relevant effects (Fig. 10a, right).

We then analyzed the effects of bradykinin on the activity of PKA, which is a key driver of the regulation of thermogenic activation in brown adipocytes[24]. Bradykinin did not alter the basal levels of PKA activity but it blunted the ability of the β3-adrenergic-activator CL316243 to induce PKA activity (Fig. 10b). We also analyzed the potential involvement of p38 MAPkinase, a main intracellular mechanism downstream of PKA that conveys thermogenic activation of brown adipocytes[25]. We found that

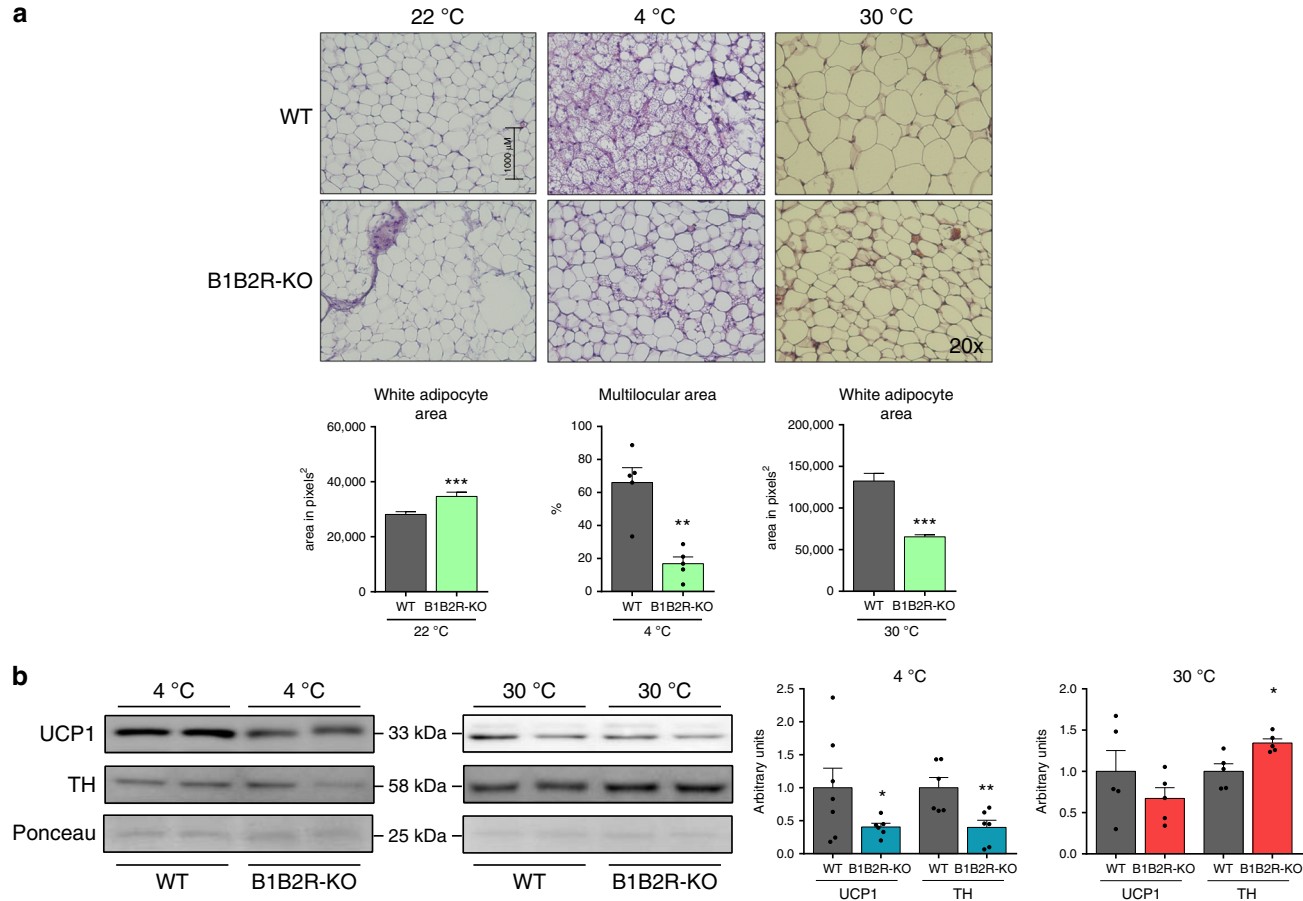

**Fig. 8 Effects of temperature challenges on iWAT of B1B2R-KO mice. a** H&E-stained histological sections and quantification (at least three different pictures) of cell size and browning area in iWAT from 3 months old WT and B1B2R-KO mice exposed to 4 °C or 30 °C for 1 week. **b** Expression and quantification of UCP1 and TH protein ($n = 6$ animals at 4 °C for UCP1 and TH except WT 4 °C UCP1 where $n = 7$; $n = 5$ at 30 °C all conditions). Data are presented as means ± s.e.m. (bars). *$P < 0.05$, **$P < 0.01$, and ***$P < 0.001$ versus corresponding controls; two-tailed unpaired Student's $t$-test. Source data are provided as a Source data file.

bradykinin reduced the p38-MAPkinase phosphorylation level and significantly impaired the induction of p38-MAPkinase phosphorylation in response to CL316243 (Fig. 10c, left) in brown adipocytes. These effects of bradykinin were suppressed in B1B2R-KO mouse-derived brown adipocytes. Other intracellular pathways potentially regulated by bradykinin in other cell systems, such as NFκB[26] (Fig. 10c, right), or Akt[27] and Stat3[28] were unaffected (Supplementary Fig. 6). These data support the idea that the PKA/p38 MAPK axis, which is the major intracellular pathway of thermogenic activation of brown adipocytes[29], is subjected to bradykinin-receptor/G$_i$-mediated repression.

## Discussion

In the present study, we identified Kng as a factor released by brown adipocytes in response to a thermogenic, noradrenergic-mediated stimulus. Although the kallikrein–kinin system is a poorly understood hormonal system, the limited available research points to a role in inflammation, blood pressure control, coagulation and pain[8]. To date, however, there has been no evidence supporting a role for this system in brown adipose tissue-related processes of adaptive energy expenditure. Here, we demonstrate that the kallikrein–kinin system is involved in the control of BAT activation and recruitment, and plays an inhibitory role in the thermogenic function of brown fat (see scheme in Fig. 10d).

We found that *Kng2* is the *Kng* gene that is preferentially expressed in BAT from mice. Expression of the *Kng2* gene and release of the encoded proteins is stimulated by a cAMP-mediated, β3-adrenergic–induced regulatory mechanism, in concert with the thermogenic activation of BAT and thermogenic-induced browning of WAT.

A strong regulatory role for the kallikrein–kinin system in BAT in the context of thermogenic stimuli is reinforced by the intense reciprocal regulation between thermogenic activation and expression of the kinin receptors B1 and B2, occurring at least at transcript levels.

Despite our observation that a noradrenergic-mediated thermogenic stimulus increased blood KNG2 levels in mice in concert with induction of *Kng2* expression specifically in BAT (and in the browning-prone iWAT depot), our various experimental interventions in the kallikrein–kinin system point to major effects in BAT itself rather than direct systemic effects. In rats defective for KNG secretion, the most remarkable feature found was upregulation of BAT activity; similarly, BAT overactivation was systematically observed in mice with pharmacological or genetic disruption of the kinin receptor system. These results are in line with our findings of inhibitory effects of Kng and bradykinin on thermogenic activation of brown adipocytes. Therefore, our findings identify a repressive action on thermogenic activity as the major effect of kallikrein–kinin signals that act upon BAT. The fact that a thermogenic stimulus (i.e., cold) results in the enhanced release of a biological signal (Kng), that exerts

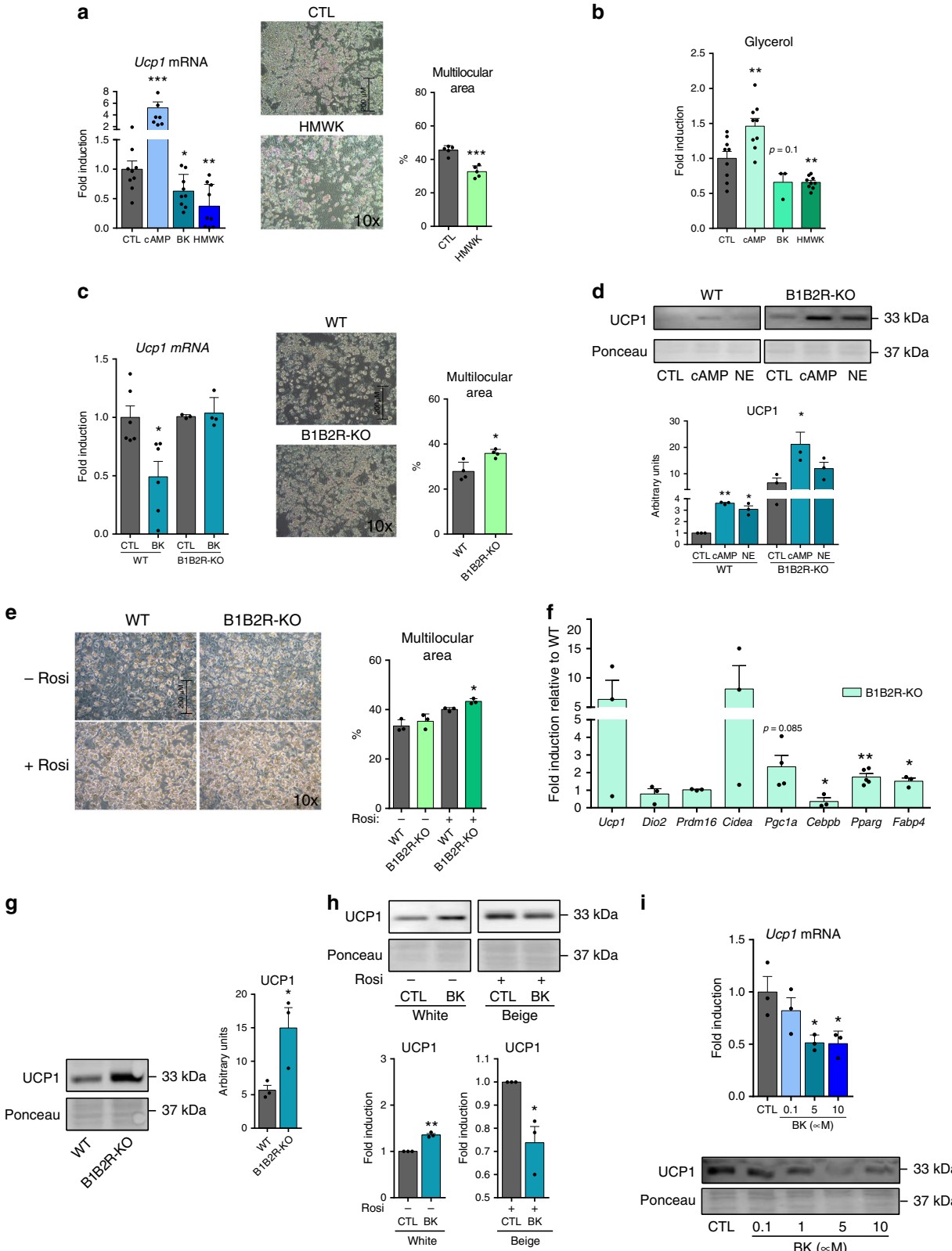

suppressive effects on thermogenic activity may seem counterintuitive at first glance, but it is not unique to the kallikrein–kinin system. Additional examples of such behavior include several brown adipocyte secreted factors such as sLR11 (a soluble relative of the low-density lipoprotein receptor)[30], endothelin A[31], endocannabinoids[32] and probably myostatin[33]. Along with these molecules, Kng may play a role in an autocrine-based

homeostatic mechanism that prevents excessive wasting of energy upon thermogenic activation, as previously proposed to explain this type of effect[30]. Although relatively few studies relating the kallikrein–kinin system with energy metabolism, it has been reported that systemic pharmacological blockade of B1 reduces adiposity and improves the metabolic profile in rats[34,35]. Moreover, a deficiency of kinin receptors in mice protects against high-

**Fig. 9 Effects of Kng and bradykinin on brown, white and beige adipocytes in culture. a** *Ucp1* mRNA expression in differentiated primary brown adipocytes incubated for 10 h with cAMP, bradykinin or recombinant HMWK protein ($n = 9$ independent cell culture experiments); representative microscopy pictures of controls (CTL) and HMWK-treated (24 h) primary brown adipocytes after 11 days of differentiation, with quantification of multilocular lipid droplets area (at least three different pictures). **b** Glycerol levels in culture media from brown adipocytes treated with cAMP, bradykinin or HMWK for 24 h ($n = 9$ independent cell culture experiments for all conditions except BK where $n = 3$). **c** *Ucp1* mRNA expression in differentiated primary brown adipocytes from WT or B1B2R-KO mice treated 24 h to bradykinin ($n = 6$ independent cell culture experiments for WT; $n = 3$ for B1B2R-KO CTL and $n = 4$ for B1B2R-KO BK); Images of WT and B1B2R-KO primary brown adipocytes after 11 days of differentiation, with quantification of multilocular lipid droplets area (at least three different pictures). **d** UCP1 protein levels in WT and B1B2R-KO primary brown adipocytes treated 24 h to different cAMP or norepinephrine (NE) ($n = 3$ independent cell culture experiments). Representative images (same immunoblot membrane, same exposure setting) of UCP1 immunoblots (top) and quantification of UCP1 levels (bottom). **e** Representative microscopy images of WT and B1B2R-KO primary white adipocytes after 8 days of differentiation in the presence or absence of the browning stimulator rosiglitazone (rosi), with quantification of multilocular lipid droplets area (at least three different pictures). **f** Fold-induction of mRNA for genes involved in thermogenesis and adipogenesis in primary white adipocytes ($n = 3$ independent cell culture experiments for all genes except for *Pgc1a* where $n = 4$ and *Pparg* where $n = 5$). **g** Expression and quantification of UCP1 protein in WT and B1B2R-KO primary white adipocytes ($n = 3$ independent cell culture experiments). **h** UCP1 protein expression and quantification of primary white ($-$rosi) and beige ($+$rosi) adipocytes after 24 h of bradykinin exposition ($n = 3$ independent cell culture experiments). **i** Expression of UCP1 mRNA and protein in primary beige adipocytes differentiated in presence of rosiglitazone after stimulation with different concentrations of bradykinin for 24 h ($n = 3$ independent cell culture experiments). Data are presented as means ± s.e.m. (bars). *$P < 0.05$, **$P < 0.01$, and ***$P < 0.001$ versus corresponding controls; *P*-values determined by one way ANOVA with Tukey's post hoc test (**a** left, **b** left, **c** left, **d**, **e**, **i**) and two-tailed unpaired Student's *t*-test (**a** right, **c** right, **f**, **g**, **h**). Source data are provided as a Source data file.

fat-diet-induced obesity and improves glucose tolerance[12,13] through mechanisms that do not involve changes in food intake, observations that are consistent with our demonstration of a repressive role of the kallikrein–kinin system on BAT activity.

In most settings, we found parallel WAT browning and BAT activation responses to kallikrein–kinin loss-of-function experimental strategies. However, in two of our experimental models— local effects of kinin receptor antagonists on WAT and response to cold in B1B2R-KO mice—blocking bradykinin action led to impaired browning in WAT. The reason for these apparently discrepant observations is unclear. In our in vitro studies, we found repressive effects due to bradykinin signaling in brown adipocytes as well as in cells induced to differentiate into beige adipocytes, but not in white adipocytes. We cannot rule out the possibility that the induction of BAT activity elicited by kinin receptor invalidation represses the extent of WAT browning owing to reciprocal compensatory processes similar to those reported in other mouse models[36,37]. However, our experimental data in cell culture point to a potential cell-autonomous differential role of the kallikrein–kinin system on WAT and BAT.

It is worth mentioning that our in vivo experimental models based on thermal challenges did not allow us to unequivocally establish whether central effects of the kallikrein–kinin system[38] are also involved in controlling of brown fat activity. Our observation that upregulation of UCP1 in models of kallikrein–kinin loss-of-function is accompanied by increased tyrosine hydroxylase, indicative of enhanced sympathetic innervation, suggest that experimental suppression of the kallikrein–kinin system elicits a global remodeling of sympathetic innervation of BAT. However, the enhanced activity of BAT in mice devoid of kinin receptors after pharmacological stimulation with a β3-adrenoceptor agonist indicates the intrinsically enhanced thermogenic capacity of BAT when kinin signaling is impaired does not rely on acute central signaling. Accordingly, we found that bradykinin can repress the thermogenic machinery of brown adipocytes in a cell-autonomous manner through its action on bradykinin receptors. Bradykinin appears to interfere with the β3-adrenergic induction of thermogenic activation in brown adipocytes through mechanisms involving the $G_i$ subtype of G-protein-coupled receptors; this results in the inhibition of protein kinase-A activity and p38-MAPkinase, which are the main canonical intracellular actors that mediate the induction of *Ucp1* gene transcription induction and overall of thermogenic activation[29]. Such a mechanism is reminiscent of some, but not

all, previous findings that have reported decreased cAMP levels in response to the activation of kinin receptors in other non-adipose cellular targets[39]. However, it has also been reported that activation of kinin receptors may either increase[40] or fail to modify[41] cAMP levels, depending on the cell type under study. Further research is warranted to explore the molecular basis of this distinct cell-specific responsiveness to bradykinin.

In summary, we have identified a previously unsuspected pathway for controlling BAT thermogenic activity mediated by the kallikrein–kinin system, which is likely to act as an autocrine, autoregulatory mechanism in response to a thermogenic stimulus. This finding may contribute to expanding the range of potential pharmacological candidates in therapeutic strategies against obesity and associated diseases designed to improve energy expenditure and remove excess blood metabolites through activation of BAT.

## Methods
**Reagents, antibodies, and primers.** All commercial kits, reagents, antibodies, primers and TaqMan probes are listed in Supplementary Methods.

**Animals.** C57BL/6-Bdkrb2/Bdkrb1[tm1Mki]/J (B1B2R-KO) mice and their littermate controls (WT) were purchased from Jackson Laboratory (Bar Harbor, ME, USA). Swiss and C57BL/6J mice, as well as Brown Norway rats (BN/RijHsd colony), were obtained from Harlan Laboratories (Indianapolis, IN, USA). BN/Ka breeder rats were kindly provided by Drs. E. Kaschina and T.Unger (Charité University, Berlin, Germany). Animals were housed at 22 °C under a 12-h light cycle (lights on, 08.00 am to 08.00 pm) with 50% ±5% relative humidity with free access to water and a standard diet.

For cold-exposure experiments, 12 weeks old B1B2R-KO, C57BL/6J, or WT Swiss male animals and their corresponding controls were exposed to a temperature of 4 °C for 1 week. For thermoneutrality experiments, 12 weeks old B1B2R-KO male animals and their controls were exposed to 30 °C for 1 week using a thermostatic chamber (TDI Euro Aire). Male, WT Brown Norway and BN/Ka rats (12 weeks old) were kept at thermoneutrality (27 °C) for 22 days or exposed to cold (4 °C) for 24 hours or 22 days. Where indicated, 9-weeks old C57BL/6J male mice were anesthetized with 1.5% isoflurane and subcutaneously implanted in their interscapular or inguinal region with an osmotic mini-pump. Mini-pumps, calibrated to release their content at a rate of 0.5 μl/h for 7 days, were filled with phosphate-buffered saline (PBS) or a cocktail of B1 (R715) and B2 (HOE 140) antagonists at a concentration designed to deliver the respective agents at a rate of 700 and 400 μg/kg/d. Animals and food were weighed daily during temperature-challenge treatments. Where indicated, mice were injected subcutaneously into the interscapular region over the BAT depot with 1 mg/kg CL316243, and their surface temperature was monitored every minute for 10 min with a T335 infrared digital thermography camera (FLIR Systems)[42–44]. The same camera was used for obtaining thermographic images of rats; images were analyzed using the FLIR QuickReport software (FLIR Systems). Oxygen consumption was

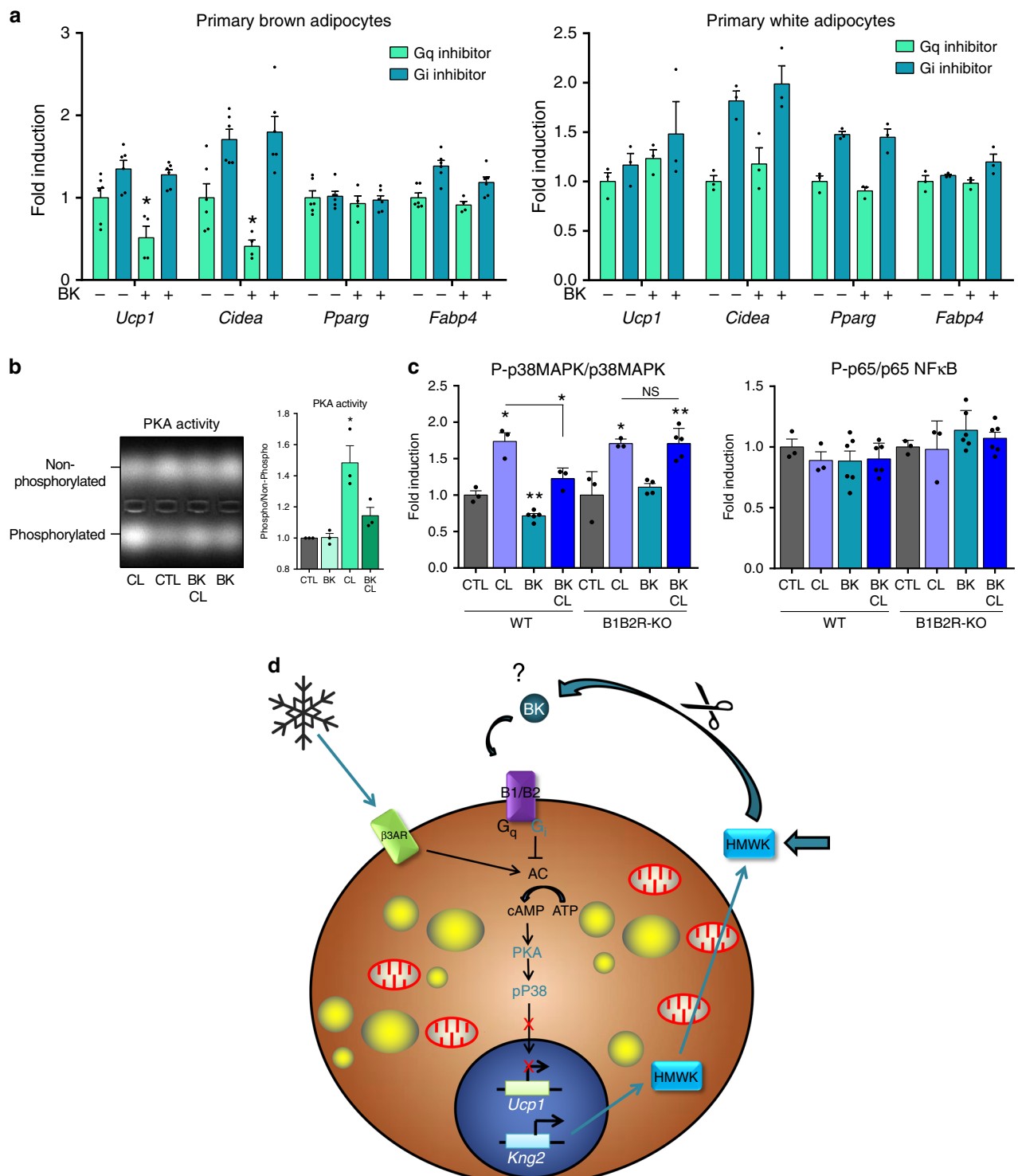

**Fig. 10 Regulation of thermogenic pathways by the kallikrein–kinin system. a** mRNA expression of genes involved in thermogenesis and adipogenesis in primary brown ($n = 6$ independent cell culture experiments, except for $G_q$/BK condition where $n = 4$) and white adipocytes ($n = 3$ independent cell culture experiments) exposed to $G_{q/i}$ inhibitors in presence or absence of bradykinin for 24 h. **b** Protein kinase A activity in primary brown adipocytes exposed to bradykinin, CL316243 or both for 15 min. Representative picture of electrophoresis assay of phosphorylated and non-phosphorylated PKA substrate and quantification ($n = 3$ independent cell culture experiments). **c** Phosphorylation status of p38MAPK ($n = 3$ independent cell culture experiments, except for WT BK where $n = 5$ and B1B2R-KO BK and BK/CL where $n = 4$ and $n = 6$, respectively) and p65 NFκB ($n = 3$ independent cell culture experiments, except for WT and B1B2R-KO BK and BK/CL where $n = 6$) in primary WT and B1B2R-KO brown adipocytes exposed to CL316243, bradykinin or both for 15 min, normalized with total p38MAPK and p65 NFκB. **d** Schematic representation of the effects of the kallikrein–kinin system in brown/beige adipocytes. Data are presented as means ± s.e.m. (bars). *$P < 0.05$ and **$P < 0.01$ versus corresponding controls; $P$-values determined by one way ANOVA with Tukey's post hoc test (**c**) and two-tailed unpaired Student's t-test (**a**, **b**). Source data are provided as a Source data file.

measured in mice housed individually in Labmaster metabolic cages (TSE, Bad Homburg, Germany) after 2 days of adaptation[45].

All experiments were performed in accordance with European Community Council directive 86/609/EEC, and experimental protocols as well as the number of animals, determined based on the expected effects size, were approved by the Institutional Animal Care and Use Committee at the University of Barcelona. After being anesthetized, 2–5 months old mice and rats were sacrificed by rapid decapitation, and blood/tissues were collected and stored at −80 °C.

**Cell culture**. Stromal vascular cells were obtained from iBAT dissected from 3-weeks old Swiss mice (mix of males and females). Differentiation was induced by exposing confluent precursor cells in DMEM/F12 medium containing 10% foetal bovine serum and supplemented with 20 nM insulin, 2 nM T3 and 0.1 mM ascorbic acid. Experiments were performed on day 10 of culture, when more than 90% of cells were differentiated, or on days 4, 6, and 8 during differentiation, as indicated. For culture of primary white adipocytes, stromal vascular cells were obtained from iWAT of 5-week old WT or B1B2R-KO mice, and induced to differentiate into adipocytes by maintaining confluent precursors in DMEM/F12 containing 10% newborn calf serum, supplemented with 850 nM insulin, 3 μM T3 and 35 nM dexamethasone. Beiging was induced by adding rosiglitazone (10 μM) to the cell media during differentiation at the time cells had reached confluence. Experiments were performed on day 8 of culture, when more than 90% of cells were differentiated. Where indicated, cells were exposed to cAMP (1 mM), nor-epinephrine (0.5 μM), CL316243 (1 μM), bradykinin (1 μM), HMWK (10 nM), $G_q$ inhibitor YM254890 (10 μM) or $G_i$ inhibitor (pertussis toxin) (100 ng/ml) at the indicated timing (24 h or 15 min) on day 10 (brown adipocytes) or day 8 (beige adipocytes) of in vitro differentiation. In the case of $G_{q/i}$ experiment, inhibitors were added 30 min before bradykinin. Microscopy images were obtained, multilocular lipid droplets areas were quantified using Image J 2.0 and cells were collected for RNA and protein extraction.

**Circulating parameters**. Triglyceride and glucose levels were measured in blood using an Accutrend Technology system (Roche Diagnostics, Basel, Switzerland). Insulin, leptin, and cytokines were measured in plasma using a Multiplex system (MADKMAG-HK, Millipore). All measurements have been performed on fed animals.

Glycerol was measured in cell culture media, using spectrophotometric methodology.

**Histological analysis**. Tissues were fixed overnight in 4% paraformaldehyde, sectioned (5-μm thick), and stained with hematoxylin and eosin (H&E) for further morphological investigation. The surface area of lipid droplets and the percentage of browning in adipose tissue areas were quantified with Image J 2.0 (NIH) and Adiposoft extension.

**Immunofluorescence**. For immunofluorescence, rehydrated tissue sections were blocked by incubating with 3% bovine serum albumen (BSA) for 1 h at room temperature. Preparations were then incubated with rabbit anti-UCP1 antibody at a dilution of 1/100, followed by incubation with AlexaFluor 488-conjugated anti-rabbit secondary antibody at a dilution of 1/1500. Immunofluorescence signals were visualized under a fluorescence microscope (Leica) and quantified using Image J 2.0.

**Real-time PCR**. Total RNA was extracted using a Nucleospin RNA column kit and reverse transcribed using a high capacity cDNA kit. Samples were systematically checked for no-amplification in the absence of reverse transcriptase. Quantitative real-time PCR was performed using a TaqMan system; in cases where TaqMan probes were not available, SYBR Green was employed in conjunction with a StepOne PCR system (Life Technologies, Carlsbad, CA, USA). Primer sequences for SYBR Green-based PCR and assay probes for TaqMan are listed in Supplementary Methods. cDNA levels for the gene of interest were normalized to that of the reference control using the comparative $C_T (2^{-\Delta\Delta CT})$ method, according to the manufacturer's instructions. A transcript was considered to be not-detectable for CT values ≥40.

**Western blot and milliplex analysis**. Homogenized tissues or cells were lysed in ice-cold RIPA buffer. Equal amounts of protein were separated by sodium dodecyl sulfate-polyacrylamide gel electrophoresis (SDS-PAGE) on 12% gradient gels and blotted onto PVDF (polyvinylidenedifluoride) membranes. Proteins were probed with specific primary antibodies (dilution 1/1000) and HRP-conjugated secondary antibodies (dilution 1/3000) in conjunction with chemiluminescence detection. Loading controls were established using Ponceau or Coomassie staining. Quantification was performed using Multi Gauge V3.0 Software (Fujifilm, Tokyo, Japan).

For milliplex multipathway analysis of total and phosphorylated proteins, cells were lysed and processed according to the manufacturer instructions (MADKMAG-HK, Millipore). The reading was done using a Luminex system.

**PKA activity**. PKA enzymatic activity was measured using the PepTag® Non-Radioactive Protein Kinase Assay kit. Cells were exposed to bradykinin, CL316243, or both, at the indicated concentrations for 15 min. Enzyme activity was estimated through separation of the non-phosphorylated versus phosphorylated PKA substrate on a 0.8% agarose gel, followed by densitometric measurement of signal intensity, and calculation of the phosphorylated/non-phosphorylated ratios, according to the manufacturer instructions.

**Statistical analysis**. Results are expressed as means ± SEM of at least three independent experiments or at least four different animals per group. Results were analyzed by Student's $t$-test or one-way analysis of variance (ANOVA) followed by a Tukey's multiple-comparisons test, in cases where more than two group groups or multiple time points, using GraphPad statistical software (GraphPad Software Inc., San Diego, CA, USA). $P$-values < 0.05 were considered significant; individual $P$-values are shown in figure legends.

**Reporting summary**. Further information on research design is available in the Nature Research Reporting Summary linked to this article.

## Data availability

RNA-seq performed for Quesada-Lopez et al. study[46] was used here. The raw data are accessible in Gene Expression Omnibus (GEO) using accession number GSE77534. The complete list of cold-modulated genes in BAT revealed by analysis of RNA-seq data is available at [http://lmedex.ulb.ac.be/data.php]. Source data underlying this study is available as a Source Data file, or from the corresponding author upon reasonable request.

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

## Acknowledgements

We thank A. Peró and M. Morales for technical support. This work has been supported by the Swiss National Science Foundation (P300PA_174342), the Ministerio de Ciencia, Innovación y Universidades y MINECO, Spain (SAF2017-85722R, RTI2018-101840-B-100 (for M.L.) and PI17/00420), co-financed by the European Regional Development Fund (ERDF) and Xunta de Galicia (ML: 2016-PG068). M.P. is currently a Juan de la Cierva researcher supported by Ministerio de Ciencia, Innovación y Universidades, Spain. T.Q.-L. was supported by a CONACyT (National Council for Science and Technology in Mexico) Ph. D. scholarship. FV is an ICREA Academia researcher.

## Author contributions

The experiments were conceived and designed by M.P. and F.V.; experiments with mice were performed by M.P., T.Q.-L., L.L.-P., and A.G.-N.; experiments with rats were performed by R.C.; cell culture experiments were performed by M.P. T.Q.-L., and L.C.; bioinformatics analysis were performed by R.C.; analysis of microscopy images was performed by M.P.; overall data were analyzed by M.P., M.G., M.L., and F.V. The paper was written by M.P. and F.V. and revised/approved by all contributors.

## Competing interests

The authors declare no competing interests.
