## [Peer Review File · Nature Communications]

Reviewers' Comments:

Reviewer #1:

Remarks to the Author:

This study addresses the effects of the kininogen/bradykinin signalling system on brown adipocyte functions. The authors show that expression of the Kng2 gene correlates with the level of thermogenic stimulation. Surprisingly, inhibition of the signalling pathway by antagonists or receptor gene inactivation markedly enhances the thermogenic response, suggesting that release of BKNs from BAT may possibly be an auto-regulatory loop. The data presented are of excellent quality and the analyses conducted up to this stage are expert and require very little to improve. This is therefore a potentially interesting study with interest to a broad readership and a good level of novelty as it identified a new pathway to regulate thermogenesis. However, the mechanistic evidence provided is to some degree a weakness of the manuscript that should be addressed more thoroughly. Specifically, the authors could consider addressing the following items:

a. As far as I can tell, there is no direct experimental link between Bradykinin or KNG2 and B1/B2 inactivation. This would be of particular interest since the observations from gain- and loss-of-function models seem to oppose each other, or else suggest a negative feedback. It would be important to demonstrate that B1B2R-KO mice or isolated cells from such animals, as shown in Figure 9, no longer respond to BK- or HWMK-treatment.

b. In relation to the previous comment: What is the mechanism of activation by BKN2 through B1/B2? The authors discuss the possibility of Gq activation, which could be easily addressed experimentally. At the very least the authors should provide western blot analysis of the most common thermogenesis regulations pathways, such as beta-adrenergic signalling, insulin signalling, and lipid metabolism in some of the in vitro or in vivo models.

c. It seems initially counterintuitive that a pro-thermogenic stimulus like cold would also increase expression of Batokines that are negative regulators of thermogenesis. On which level does Bkn2 act to inhibit thermogenesis or do brown (but not white) adipocytes become kininogen/bradykinin-insensitive during thermogenic activation? This would also help to shed light onto the apparent discrepancies the authors discuss when comparing BAT/WAT and in vitro/in vivo.

d. The negative effects of the natural ligand, BK, have only been addressed on the level of UCP1 expression. In order to determine whether kininogen/bradykinin do impair thermogenesis cell-autonomously, functional assays, such as lipolysis rates and mitochondrial function should be analysed. Moreover, it would be helpful to determine whether there is an interaction with the adrenergic system, which some data suggest. To this end, data shown in Figure 9A and 9D should also be measured in cells exposed to cAMP or the adrenergic agonist.

e. It is not clear to me what is shown in Figure 5B. Is this UCP1 Protein or all total protein in BAT? Please indicate in figure legend.

Reviewer #2:

Remarks to the Author:

The work from Marion Peirou et al entitled "The kininogen/bradykinin pathway, a mechanism for autocontrol of brown adipose tissue activity" brings a new message regarding the role of the kallikrein-kinin system (KKS) as a relevant component of brown adipose tissue (BAT) thermogenic regulation. The authors have convincingly demonstrated that the KKS might be regarded as a novel, previously unsuspected pathway for controlling BAT thermogenic regulation, acting as an autocrine, autoregulatory mechanism in response to a thermogenic stimulus. As pointed out by the authors, this finding may contribute to expanding the range of potential pharmacological candidates in therapeutic strategies against obesity and associated diseases designed to improve

energy expenditure. This is a novel, original finding, which will be of interest to the wide community of researchers and clinicians.

The work is well done but this reviewer feels that some flaws are evident in the manuscript and should be pointed out, as follows:

1) Introduction:

- Please refer to the system as kallikrein-kinin and not as kininogen-bradykinin system.
- Please also use the correct nomenclature of the receptors as kinin B1 or kinin B2 receptors. Bradykinin only binds to the kinin B2 receptor and therefore the receptor should be named kinin B2 (or bradykinin B2 receptor). However the B1 receptor binds the des-Arg agonists [des-Arg9]-bradykinin/[des-Arg10]-kallidin and not bradykinin. Therefore it should be named kinin B1 receptor.
- The agonists [des-Arg9]-bradykinin/[des-Arg10]-kallidin are not the final peptides from the kinin pathway, as described in the text.

2) Results:

- Sometimes the authors normalize protein expression in Western blots with Ponceau, sometimes with Coomassie or beta-actin. Why these different normalizers?
- Why there are two different bands for KNG2 in WBs? Sometimes the upper band is the stronger one and sometimes the lower band. Why does this happen? Are these antibodies reliable?
- How the authors explain the positive response to thermogenic stimulus at the RNA level resulting from the alanine to threonine mutation at residue 163 in the KNG2 gene in the Brown Norway Katholiek rats?
- Regarding the down/up regulation in the kinin receptors depending on the temperature, the authors should show the effect also at the protein level. The authors did not provide a rationale for this effect. Is the system activated in this condition? Is there an increased production of kinins in these conditions? Is the activity of the other components of the system (kallikrein, kininases) modulated?
- This reviewer wonders why the authors decide to study the double knockout mice instead of the individual B1 and B2 knockout mice. The same question is valid for the use of antagonists: why using both instead of blocking separately each receptor? Since both receptors have different pharmacology it is important to pinpoint the role of each one in the thermogenic regulation in BAT.
- Morais et al (reference 17) show that the double B1B2R KO mice present normal glucose levels but lower levels of the hormones leptin and insulin at normal conditions, which is in agreement with the data published for the single kinin receptor knockouts. As long as there are two different strains of double KO generated at different laboratories, how do the authors explain this difference? How does this data impact the findings presented in this manuscript? This discrepancy should be discussed in the present manuscript.
- The authors should provide details of protocol used for the Real-Time PCR for the kinin receptors. The primers and probes supplied by the manufacturer map within a single exon and should also detect genomic DNA and could interfere with the data presented. This assumption should be valid for the other assays used for the other genes. Therefore the protein expression/functional data should be always performed.

-

Reviewer #3:

Remarks to the Author:

The paper by Peyrou describes the analysis of the Kininogen/Bradykinin system in the regulation of brown fat activity. Through the use of several models, both in vitro and in vivo they show that BAT expresses HMWK and LMWK, furthermore they demonstrate that especially in BAT signaling through B1 and B2 receptor modulates the activity of this tissue. This is an interesting finding which put another regulator of BAT on the map which seems to act independent of the classical NE mediated pathways. However there are several points that need to be addressed:

1. Fig. 1. Please show the absolute levels of HMWK and LMWK and not just inductions this is

especially important in relation to the question where circulating HMWK is coming from (see below). Furthermore, a measurement of Kng2 and 1 in tissues should be performed to elucidate whether the induction is happening on a protein level. Again, this is required to draw any conclusion about the origin of the circulating factors. For Fig. 1D please provide the individual data points.

2. Fig. 2. How do the authors explain the differences in the LMWK changes seen in vivo and in vitro. Please also provide a quantification of the blots 2B and 2C (similar to 5C).

3. Fig. 3. Please show 3E at full scale.

4. I am extremely puzzled by the data in Fig. 6. At 4C Ucp1 is slightly increased in B1/B2 ko mice and the effect is much more pronounced at TN. Why then do the authors not observe an effect under mild cold stress ?

5. Fig. 9. A much more careful analysis of B1/B2 cell autonomous differentiation is required. In my opinion the two images show a difference and PPARg levels (even though it is difficult to see seems to be upregulated by at least 50%).

6. The cartoon presented in Fig. 9 is in my opinion an overinterpretation. From the current data it is difficult to determine whether the HMWK is indeed acting in a cell autonomous manner, especially in vivo. To show this the authors would need to do ko experiments in cultured brown adipocytes coupled with media transfer to demonstrate that indeed HMWK acts in an autocrine or paracrine manner.

7. The finding of differential effects on WAT vs BAT browning is very intriguing especially since the regulation of LMWK seems to be conserved between the two tissues. Could this be due to different modes of induction (i.e de novo recruitment of beige cells vs. activation of existing cells ?). This should be discussed in more detail.

Reviewer #1 (Remarks to the Author):

This study addresses the effects of the kininogen/bradykinin signalling system on brown adipocyte functions. The authors show that expression of the *Kng2* gene correlates with the level of thermogenic stimulation. Surprisingly, inhibition of the signalling pathway by antagonists or receptor gene inactivation markedly enhances the thermogenic response, suggesting that release of BKNs from BAT may possibly be an auto-regulatory loop. The data presented are of excellent quality and the analyses conducted up to this stage are expert and require very little to improve. This is therefore a potentially interesting study with interest to a broad readership and a good level of novelty as it identified a new pathway to regulate thermogenesis. However, the mechanistic evidence provided is to some degree a weakness of the manuscript that should be addressed more thoroughly. Specifically, the authors could consider addressing the following items:

a. As far as I can tell, there is no direct experimental link between Bradykinin or KNG2 and B1/B2 inactivation. This would be of particular interest since the observations from gain- and loss-of-function models seem to oppose each other, or else suggest a negative feedback. It would be important to demonstrate that B1B2R-KO mice or isolated cells from such animals, as shown in Figure 9, no longer respond to BK- or HWMK-treatment.

In accordance with the reviewer suggestion we checked whether the effect of bradykinin was lost in B1/B2R-KO cells. Figure 9C in the revised manuscript shows that bradykinin is unable to down-regulate *Ucp1* mRNA expression in primary brown adipocytes from B1B2R-KO mice, in contrast with the effect in wild-type cells. Moreover, the B1B2R-KO brown adipocytes were unresponsive to the bradykinin-induced p38-MAPkinase down-regulation that occurred in wild-type cells in response to bradykinin (Fig. 10C in the revised manuscript).

b. In relation to the previous comment: What is the mechanism of activation by BKN2 through B1/B2? The authors discuss the possibility of Gq activation, which could be easily addressed experimentally. At the very least the authors should provide western blot analysis of the most common thermogenesis regulations pathways, such as beta-adrenergic signalling, insulin signalling, and lipid metabolism in some of the in vitro or in vivo models.

We expanded our analysis of the pathways downstream of the bradykinin-triggered repression of thermogenic activation in brown adipocytes.

We analyzed the potential involvement of G_q- and G_i-coupled receptor-mediated signaling in the ability of bradykinin to repress *Ucp1* gene expression (as an indicator of the overall repressive effects of bradykinin on thermogenic activation). Fig 10A (revised manuscript) present results related to the effects of bradykinin when the G_q or G_i-mediated pathway was blocked using specific inhibitors. Our experiments consistently showed that G_i is required for the ability of bradykinin to repress thermogenic gene expression in brown adipocytes.

We performed PKA activity assays (examining the classical intracellular mechanism that drives the initial steps of β-adrenergic stimulus-induced thermogenic activation) and we found that bradykinin repressed β3 adrenergic-induced PKA activity (Fig. 10B in the revised manuscript).

We explored the potential effects of bradykinin in intracellular phosphorylation signaling events known to be involved in thermogenic activation or bradykinin-mediated signaling in other cell types. We found that bradykinin impairs the phosphorylation of p38-MAPkinase, which is a well-known inducer of the thermogenic program downstream of PKA in brown adipocytes. We did not find significant bradykinin-induced changes in any of the other tested pathways; these included NFκB (which was reported to be altered by bradykinin in other cells systems, ref. 26), phospho-AKT (which is a key component of insulin signaling and also involved in bradykinin action in non-adipose cell systems, ref 27) and phospho-STAT3 (also reported as involved in bradykinin action in several cell systems, ref 28). (See Fig. 10C and Supplementary Fig. 6 in the revised manuscript).

These findings related to bradykinin-induced signaling are now presented in new Fig. 10 of the revised manuscript.

We also expanded our analysis of the targeting of bradykinin to the measurement of lipolysis, which is a key process related to the thermogenic activation of brown adipocytes. We observed that bradykinin inhibited lipolysis, which is concordant with overall inhibitory effects of bradykinin on thermogenic activation in brown adipocytes (see Fig. 9B of the revised manuscript).

c. It seems initially counterintuitive that a pro-thermogenic stimulus like cold would also increase expression of Batokines that are negative regulators of thermogenesis. On which level does Bkn2 act to inhibit thermogenesis or do brown (but not white) adipocytes become kininogen/bradykinin-insensitive during thermogenic activation? This would also help to shed

light onto the apparent discrepancies the authors discuss when comparing BAT/WAT and in vitro/in vivo.

Effectively, it could seem counterintuitive that a batokine induced after thermogenic activation of BAT may negatively regulate thermogenic activation. However, as stated in the Discussion, there are several previous examples of such behavior (sLR11, endothelin A, endocannabinoids, myostatin). The existence of this panel of thermogenesis-induced secreted factors with autocrine repressive effects on thermogenesis suggests that these negative feedback-based mechanisms of auto-control may play a relevant physiological role. We have expanded somewhat on these points in the Discussion section of the revised manuscript.

We also further analyzed the apparent discrepancy between the ability of bradykinin to repress brown adipose tissue thermogenic activation and its opposite (or nonexistent) effect in white adipocytes. We found that this difference actually occurs in a cell-autonomous manner (Fig. 9 in the revised manuscript). White adipocytes differentiated in culture responded oppositely to bradykinin (induction of UCP1) in comparison with brown adipocytes (repression of UCP1). However, when cultures of white adipocyte precursors were induced to undergo browning (rosiglitazone treatment), bradykinin caused a reduction in UCP1 protein levels, as seen in brown adipocytes (see Fig. 9H in the revised manuscript).

d. The negative effects of the natural ligand, BK, have only been addressed on the level of UCP1 expression. In order to determine whether kininogen/bradykinin do impair thermogenesis cell-autonomously, functional assays, such as lipolysis rates and mitochondrial function should be analysed. Moreover, it would be helpful to determine whether there is an interaction with the adrenergic system, which some data suggest. To this end, data shown in Figure 9A and 9D should also be measured in cells exposed to cAMP or the adrenergic agonist.

We expanded our analysis of the targeting of bradykinin to the measurement of lipolysis, which is a key process related to thermogenic activation of brown adipocytes. We observed that bradykinin inhibited lipolysis, which is concordant with the overall inhibitory effects of bradykinin on thermogenic activation in brown adipocytes (see Fig. 9B).

Concerning the interaction of bradykinin with the adrenergic system we found that bradykinin had significant repressive effects on β 3-adrenergic agonist (CL)-induced PKA and p38 MAPkinase activities (Fig. 10). These data, together with our findings related to the involvement of G_i , suggest that bradykinin acts downstream of adrenergic signaling by down-regulating the

capacity of the β 3-adrenergic activation to induce the pKA/p38-MAPK axis of thermogenic activation.

e. It is not clear to me what is shown in Figure 5B. Is this UCP1 Protein or all total protein in BAT? Please indicate in figure legend.

It is clarified that Figure 5B corresponds to iBAT total protein. This is shown as an indicator of overall recruitment of potentially “active” BAT tissue, to avoid misleading interpretations based on iBAT weight in relation to functional tissue recruitment as potentially affected by variations in fat content.

Reviewer #2 (Remarks to the Author):

The work from Marion Peirou et al entitled “The kininogen/bradykinin pathway, a mechanism for autocontrol of brown adipose tissue activity” brings a new message regarding the role of the kallikrein-kinin system (KKS) as a relevant component of brown adipose tissue (BAT) thermogenic regulation. The authors have convincingly demonstrated that the KKS might be regarded as a novel, previously unsuspected pathway for controlling BAT thermogenic regulation, acting as an autocrine, autoregulatory mechanism in response to a thermogenic stimulus. As pointed out by the authors, this finding may contribute to expanding the range of potential pharmacological candidates in therapeutic strategies against obesity and associated diseases designed to improve energy expenditure. This is a novel, original finding, which will be of interest to the wide community of researchers and clinicians.

The work is well done but this reviewer feels that some flaws are evident in the manuscript and should be pointed out, as follows:

1) Introduction:

- Please refer to the system as kallikrein-kinin and not as kininogen-bradykinin system.

We have made the requested change throughout the manuscript

- Please also use the correct nomenclature of the receptors as kinin B1 or kinin B2 receptors. Bradykinin only binds to the kinin B2 receptor and therefore the receptor should be named kinin B2 (or bradykinin B2 receptor). However the B1 receptor binds the des-Arg agonists [des-Arg9]-bradykinin/[des-Arg10]-kallidin and not bradykinin. Therefore it should be named kinin B1 receptor.

We have corrected the nomenclature to the more precise terminology indicated by the reviewer. Although “bradykinin receptors” has been often used to refer to the kinin B1 and B2 receptors in the literature, we agree that it is better to be precise in this regard.

The agonists [des-Arg9]-bradykinin/[des-Arg10]-kallidin are not the final peptides from the kinin pathway, as described in the text.

We appreciate this comment, and have made the necessary correction.

2) Results:

- Sometimes the authors normalize protein expression in Western blots with Ponceau, sometimes with Coomassie or beta-actin. Why these different normalizers?

We initially employed β -actin for normalization of our immunoblots, but we realized that in some of our experimental settings (especially our “*in vivo*” studies) β -actin exhibited variations that were not compatible with its use as a control. Therefore, we moved to using overall staining of the blots as a loading control. Most often we employed Ponceau staining instead of Coomassie, as this allowed us to re-probe the Western blots if needed.

- Why there are two different bands for KNG2 in WBs? Sometimes the upper band is the stronger one and sometimes the lower band. Why does this happen? Are these antibodies reliable?

The antibody used against KNG2 recognizes a region common to HMWK isoform 1 and HMWK isoform 2, which are very close size. Therefore, we expected to observe a double band in the plasma. The size of KNG2 was mislabeled in the Western blot image presented in Figure 1 in the original version of the manuscript. We apologize for this error, and have corrected it in the revised Figure.

In a single animal, the lower band (KNG1) was more intense than the upper band, perhaps due to inter-animal variability. For that reason, quantifications were performed in at least 3-4 animals per group.

Regarding antibodies, the antibody against KNG1 is commercial (see Supplementary Methods), and has been used in previously published studies (e.g., Wu et al. Int J Mol Med 17, 2016). The antibody recognized mostly two bands with sizes compatible with those predicted for HMWK.

We designed the KNG2 antibody against the peptide “lyrvtkrakmdgsat” corresponding to an amino acid sequence in the KNG2 highly divergent from KNG1. The antibody detected a band of the expected size in immunoblots. Moreover, there was a high concordance between differences in band intensity variation found among tissues and experimental conditions when we compared the protein (using the KNG2 antibody) and Kng2 transcript levels in the same tissues and cell samples.

- How the authors explain the positive response to thermogenic stimulus at the RNA level resulting from the alanine to threonine mutation at residue 163 in the KNG2 gene in the Brown Norway Katholiek rats?

In this rat model, the mentioned mutation results in the near absence of circulating KNG2 in blood due to defects in its secretion by tissues (including brown fat). The effects of KNG2 (or derived bradykinin) in brown fat are expected to be due to their actions from the extracellular compartment of brown adipocyte cells, either because endocrine (exposure of brown adipocytes to systemic KNG2) or autocrine (exposure to brown adipocytes to secreted KNG2) actions. Therefore, when brown adipocytes experience decreased exposure to extracellular KNG2, they lose its “repressive” effects on thermogenic activation. This leads to the observed over-activation of BAT seen in the BNK rat model.

- Regarding the down/up regulation in the kinin receptors depending on the temperature, the authors should show the effect also at the protein level. The authors did not provide a rationale for this effect. Is the system activated in this condition? Is there an increased production of kinins in these conditions? Is the activity of the other components of the system (kallikrein, kininases) modulated?

We tried to provide data on protein levels for kinin receptors using commercially available antibodies (B1, Invitrogen PA5-77292; B2, Thermofisher 720288). However, we could not retrieve reliable data in immunoblots, especially when validated used B1B2-KO (no expression of the B1 and B2 receptor transcripts) samples. There are a few publications in the literature reporting immunoblots on B1 and B2 receptors in cells other than adipocytes, and using other antibodies. It is possible to undertake an exhaustive testing of other commercially available antibodies if there is an absolute editorial requirement for provision of this immunoblot-based information for publication. However, we think that obtaining reliable data by expanding the number of antibodies to be tested is not guaranteed, in light of existing literature. We think that,

in the current stage of this research field, our observation of changes in B1 and B2 receptor expression in response to environment temperature based on transcript levels is of interest, but not strictly associated with the core message of our manuscript. So, we prefer to maintain the data as shown, with explicit statement of the limitations of showing data only on transcript levels (see Discussion).

We present evidence indicating that the induction of the KNG agonist system (KNG2 expression and secretion) in response to cold is accompanied by a reciprocal repression of the expression of receptors (at least, at transcript level) that appear to be involved in the pathway. This sort of signaling is not unusual in BAT regulation. For example, induction of β 3-adrenergic mediated thermogenesis is accompanied by a concomitant repression of β 3 receptor expression (Bengtsson et al. J Biol Chem. 1996, 271, 33366). The nuclear receptor PPAR γ is down-regulated upon activation of thermogenesis despite its role in mediating positive effects in the thermogenic pathway (Lindgren et al. Biochem J. 2004, 382, 597). We speculate that this behavior is part of a homeostatic system that acts against an excessive and sustained activation of a regulatory pathway that requires fine tuning of its up- and down-regulation, but not sustained activity.

As requested, we measured KNG2 protein levels in BAT and WAT. Our results verify that the KNG2 protein levels are up-regulated in response to cold (see Fig. 1C in the revised manuscript).

We measured the gene expression levels of other components of the kinin system, including plasma kallikrein b1 (Klk1), kininase I (Klk1), and angiotensin-converting enzyme (Ace) in BAT and WAT. The levels of expression of transcripts for plasma kallikrein b1 (Klk1) and kininase I (Klk1) were very low in BAT and WAT (around 35 CTs in the qRT-PCR assays run using standard procedures; in fact, Klk1 mRNA was not detected with CT > 40 in BAT). Unlike to our findings for KNG2 and the kinin receptors, these transcripts did not appear to exhibit any consistent environmental temperature-dependent regulation (Supp Fig.3 in the revised version).

This reviewer wonders why the authors decide to study the double knockout mice instead of the individual B1 and B2 knockout mice. The same question is valid for the use of antagonists: why using both instead of blocking separately each receptor? Since both receptors have different pharmacology it is important to pinpoint the role of each one in the thermogenic regulation in BAT.

Since our study is just the first to report the role of the overall kallikrein/kinin system in brown adipose tissue and adipose plasticity, we wanted to ensure our “loss-of-function” model. This is why we chose the double KO model for B1 and B2. As our research was oriented by exploring the role of kininogen, we couldn't *a priori* rule out the possibility that distinct kininogen-derived products (e.g. kallidin) may act upon B1 and B2 in our system (Regoli et al., Pharmacol Rev. 1980 Mar;32, 1) acting upon B1 and B2. Other authors who also used double B1B2R-KOs (Morais et al. ref 17) noted that in some experimental settings, the absence of one receptor may lead the other to be overexpressed and mediate functional compensation (Duka et al.. Circ Res. 2001;88:275; Rodrigues et al. Peptides. 2013;42:1). In the future, it will be interesting to explore the individual role of each receptor and identify their relative roles in kininogen-2 and bradykinin signaling of BAT. However, this will require the development of novel experimental tools and is beyond the goals of this first analysis of the overall behavior of the kallikrein/kinin system in BAT.

- Morais et al (reference 17) show that the double B1B2R KO mice present normal glucose levels but lower levels of the hormones leptin and insulin at normal conditions, which is in agreement with the data published for the single kinin receptor knockouts. As long as there are two different strains of double KO generated at different laboratories, how do the authors explain this difference? How does this data impact the findings presented in this manuscript? This discrepancy should be discussed in the present manuscript.

The strain of B1B2R-KO strain that we used was obtained from Jackson Laboratory and originally developed by Kakoki et al (PNAS, 2007). Our findings of unaltered body weight and glycemia are concordant with previous data obtained using this mouse model (Kakoki et al., PNAS 2010, 107:10190; Wende et al. Endocrinology 2010, 151: 3536), and are also consistent with the finding of unaltered leptinemia. It is unclear why results differ from those obtained using the other independently obtained B1 and B2 receptor-KO mouse model (Morais et al. Diabetes Metab Syndr Obes. 2015; 8: 399), which showed decreased body weight, and low levels of leptin and insulin. In principle, the genetic background (C57NL/6) is the same and there did not appear to be any major difference in the age of the studied mice. It is worth mentioning, however, that previous reports involving both strains failed to clearly detail the feeding conditions of the mice that were studied for their glucose and leptin levels. This discrepancy is mentioned in the Discussion of the revised manuscript. We also note that the data we obtained regarding adipose plasticity in the B1B2R-KO mouse were consistent with those we obtained from independent experimental approaches *in vivo* (use of B1+B2 drug antagonists) and *in vitro* (brown and white

adipocyte cultures), which reinforced our consideration of the reliability of our B1B2-KO receptor model.

- The authors should provide details of protocol used for the Real-Time PCR for the kinin receptors. The primers and probes supplied by the manufacturer map within a single exon and should also detect genomic DNA and could interfere with the data presented. This assumption should be valid for the other assays used for the other genes. Therefore the protein expression/functional data should be always performed.

We extracted RNA using an affinity-based column kit (Macherey-Nagel, Düren, Germany) that includes a step to eliminate any possible contamination with genomic DNA. The quality of the obtained RNA obtained was confirmed using NanoDrop spectrophotometer from (ThermoFisher). Samples were systematically checked to rule out any amplification in the absence of reverse transcriptase, which ensured that there was no amplification due to remnant DNA in the RNA preparations, even if using primers and probes not placed at distinct exons of the target gene. We have added this information to our description of qRT-PCR methodology.

Reviewer #3 (Remarks to the Author):

The paper by Peyrou describes the analysis of the Kinoinogen/Bradykinin system in the regulation of brown fat activity. Through the use of several models, both in vitro and in vivo they show that BAT expresses HMWK and LMWK, furthermore they demonstrate that especially in BAT signaling through B1 and B2 receptor modulates the activity of this tissue. This is an interesting finding which put another regulator of BAT on the map which seems to act independent of the classical NE mediated pathways. However there are several points that need to be addressed:

1. Fig. 1. Please show the absolute levels of HMWK and LMWK and not just inductions this is especially important in relation to the question where circulating HMWK is coming from (see below). Furthermore, a measurement of Kng2 and 1 in tissues should be performed to elucidate whether the induction is happening on a protein level. Again, this is required to draw any conclusion about the origin of the circulating factors. For Fig. 1D please provide the individual data points.

We modified Figure 1 to include the items requested by the reviewer. Specifically, we now show the absolute levels of HMWK and LMWK transcripts in Figure 1B, instead of their fold induction.

We also provide data on KNG2 protein levels in BAT and iWAT under basal and cold-induced conditions (Fig. 1C), which confirmed a cold-induced increase. It should be kept in mind, however, that measuring the steady-state level of a secreted protein such as KNG2 inside the tissue may underestimate the actual amount of protein produced because the protein undergoes active secretion once it is synthesized inside the cell. As requested, we now show the data in Fig 1D as individual points.

2. Fig. 2. How do the authors explain the differences in the LMWK changes seen *in vivo* and *in vitro*. Please also provide a quantification of the blots 2B and 2C (similar to 5C).

The data provided in Fig. 2 are intended to establish the extent to which observations in the adipose tissue (which contains adipocytes along with other cell types potentially expressing LMWK)-based observations shown in Figure 1 may reflect events occurring in adipocytes. We cannot rule out the possibility that the cold-induced induction of LMWK found in WAT *in vivo* was due to the effects of cold on cell types other than adipocytes. Another possible explanation is that NE-induced, cAMP-mediated, effects could account for most of, but not all, of the observed effects of cold (especially in BAT). Agents and pathways other than noradrenergic stimulation may be involved in cold-induced LMWK expression in WAT, and these pathways would not be mimicked by the action of NE or cAMP in cell culture.

The requested quantification of blots is now presented in Figure 2B and 2C of the revised manuscript.

3. Fig. 3. Please show 3E at full scale.

The requested change has been made to Figure 3E in the revised manuscript.

4. I am extremely puzzled by the data in Fig. 6. At 4C Ucp1 is slightly increased in B1/B2 ko mice and the effect is much more pronounced at TN. Why then do the authors not observe an effect under mild cold stress ?

The reviewer's point is highly pertinent. It is also consistent with our rationale that the KNG system is particularly important for adaptive adipose tissue plasticity in response to thermal challenge, but it may be less important in determining the long-term status of mice across a lifespan spent under a moderate and sustained cold challenge. We would like to stress out that, mice were raised and maintained at 22°C and our experimental setting was designed to cause a

rapid adaptation (1 week) to enhance thermogenesis (cold) or to suppress it (thermoneutrality). The response of our loss-of-function model to this manipulation allowed us to identify the importance of KNG2 in this acutely required adaptive responses.

5. Fig. 9. A much more careful analysis of B1/B2 cell autonomous differentiation is required. In my opinion the two images show a difference and PPAR γ levels (even though it is difficult to see seems to be upregulated by at least 50%).

We agree with the reviewer. To address this issue, we analyzed several additional independent differentiation experiments and quantified the extent of differentiation (the surface of culture wells occupied by lipid-containing cells). Our data confirmed a moderate but statistically significant induction of differentiation in B1B2R-KO brown adipocytes (Fig. 9C in the revised manuscript). We also expanded the number of independent samples for gene expression analyses and our data confirm that *Ppar γ* and *Fabp4* were significantly up-regulated in beige adipocytes from B1B2KO mice. This indicated that the absence of B1/B2 receptor led to an over-induction of differentiation. The manuscript has been modified accordingly (see Fig 9 in the revised manuscript).

6. The cartoon presented in Fig. 9 is in my opinion an overinterpretation. From the current data it is difficult to determine whether the HMWK is indeed acting in a cell autonomous manner, especially in vivo. To show this the authors would need to do ko experiments in cultured brown adipocytes coupled with media transfer to demonstrate that indeed HMWK acts in an autocrine or paracrine manner.

We modified the final cartoon, which we believe readers will need due to the complexity of the system. We added a question mark in relation to the point mentioned by the reviewer, as we agree that an autocrine effect of HMWK is likely but not fully demonstrated. We added an extra "arrow" to point out the possibility that systemic HMWK may also act on brown adipocytes. We also modified the cartoon to include information regarding intracellular signaling, which was obtained from the additional experiments performed during the revision process.

7. The finding of differential effects on WAT vs BAT browning is very intriguing especially since the regulation of LMWK seems to be conserved between the two tissues. Could this be due to different modes of induction (i.e de novo recruitment of beige cells vs. activation of existing cells?). This should be discussed in more detail.

We agree that the kallikrein-kinin system may have different effects in existing brown (or even beige) cells versus those occurring across the recruitment process to achieve the browning of WAT. In the revised manuscript, we provide additional support for differential effects of bradykinin in distinct types of adipose cells. We found that, when precursor cells from white adipose depots were induced to differentiate into a beige phenotype (rosiglitazone exposure, high *Ucp1* gene induction) bradykinin maintained a repressive effect on UCP1 expression (as seen in classical brown adipocytes). However, when precursors are maintained in a medium that did not promote the beige phenotype (absence of rosiglitazone, low *Ucp1* expression), the repressive actions of BK were lost or potentially reversed (UCP1 induction) (Fig. 9H in the revised manuscript). These findings are consistent with some of our *in vivo* data regarding the behavior of BAT and WAT in response to gain- or loss-of-function models for bradykinin-related signaling. The differential actions of BK in white and brown adipocytes may reflect differences in how the proposed G_q - and G_i -dependent couplings function in bradykinin-induced intracellular signaling. For example, whereas G_i inhibition suppressed the ability of bradykinin to repress *Ucp1* gene expression in brown adipocytes (Fig. 10A, left in the revised manuscript), no such effect was seen in white adipocytes (Fig. 10A, right in the revised manuscript). As requested by the reviewer, we have expanded the discussion to include this point within the space limitation of the journal.

Reviewers' Comments:

Reviewer #1:

Remarks to the Author:

The authors have addressed all my comments and concerns sufficiently well, and the manuscript has been improved significantly.

Reviewer #3:

Remarks to the Author:

the authors addressed all my concerns, I would suggest to accept the paper for publications

Point by point response of referee requests:

REVIEWERS' COMMENTS:

Reviewer #1:

The authors have addressed all my comments and concerns sufficiently well, and the manuscript has been improved significantly.

Reviewer #3:

The authors addressed all my concerns, I would suggest to accept the paper for publications.

We thank the reviewers for their comments that helped improving our manuscript greatly.

Reviewer #2

Please note that we were unable to receive comments from Reviewer 2. Editorially we find your response to this referee suitable and in the interest of time decided to proceed without their feedback. We would only ask you that you experimentally respond to the reviewer's point regarding antibody validation, e.g. by in vitro KD or similar approach.

We thank the editorial board for their decision.

Concerning the experimental point mentioned, we are keen to undertake further experimental assays of validation, as requested. However, KD analysis appears currently unfeasible in a reasonable time frame. The current sanitary situation in Europe and particularly in Spain has blocked experimental laboratory capacities in the following weeks, and even after checking several international companies that provide KD determination upon order, using the standard Biacore methodologies, we couldn't retrieve any reliable time frame to obtain this specific parameter.

In order to reinforce the data regarding the reliability of our antibody, additional to the already mentioned in our previous reply, we added a Supplementary Figure (Supp Fig7) showing the Western blot tests that we did during the production and purification of the antibody. It shows how purification of the antiserum, based on affinity for the specific KNG2 peptide, results in the elimination of non-specific bands and the only remaining

band with a size corresponding to KNG2 protein, and tissue expression distribution equal to *Knj2* and distinct from *Knj1*.

Given the circumstances and methodological data provided so far (in fact, more extensive than those provided for many commercial antibodies) regarding KNG2 antibody, added to the fact that the use of this antibody refers to a limited section of the manuscript, we hope that you will be able to proceed to the publication of our article.